# The plate-to-rod transition in trabecular bone loss is elusive

A. A. Felder[1,2], S. Monzem[1,3], R. De Souza[3], B. Javaheri[1,4], D. Mills[5], A. Boyde[5] and M. Doube[1,6]

[1]Royal Veterinary College, London, UK
[2]University College London, London, UK
[3]Universidade Federal de Mato Grosso, Cuiabá, Brazil
[4]City University of London, London, UK
[5]Queen Mary University of London, London, UK
[6]City University of Hong Kong, Kowloon, Hong Kong, Hong Kong Special Administrative Region of the People's Republic of China

AAF, 0000-0003-3510-9906; BJ, 0000-0001-7941-3104; MD, 0000-0002-8021-8127

**Subject Areas:**
physiology

**Keywords:**
ellipsoid factor, structure model index, plates, rods, trabecular bone, osteoporosis

**Author for correspondence:**
A. A. Felder
e-mail: a.felder@ucl.ac.uk

Changes in trabecular micro-architecture are key to our understanding of osteoporosis. Previous work focusing on structure model index (SMI) measurements have concluded that disease progression entails a shift from plates to rods in trabecular bone, but SMI is heavily biased by bone volume fraction. As an alternative to SMI, we proposed the ellipsoid factor (EF) as a continuous measure of local trabecular shape between plate-like and rod-like extremes. We investigated the relationship between EF distributions, SMI and bone volume fraction of the trabecular geometry in a murine model of disuse osteoporosis as well as from human vertebrae of differing bone volume fraction. We observed a moderate shift in EF median (at later disease stages in mouse tibia) and EF mode (in the vertebral samples with low bone volume fraction) towards a more rod-like geometry, but not in EF maximum and minimum. These results support the notion that the plate to rod transition does not coincide with the onset of bone loss and is considerably more moderate, when it does occur, than SMI suggests. A variety of local shapes not straightforward to categorize as rod or plate exist in all our trabecular bone samples.

## 1. Introduction

The metabolic bone disease osteoporosis is a major health concern associated with high mortality rates and considerable economic costs [1,2], likely to be exacerbated by the increase in the proportion of elderly people in future demographics. In this disease, imbalance between osteoblastic (bone-forming) and osteoclastic (bone-resorbing) cell activity is thought to lead to lower bone turnover and relatively higher resorption than

formation, and thus to a lower amount of bone [3]. Lower bone mass causes reduced mechanical competence and increased fracture risk with age [4].

Bone tissue is typically classified by porosity into trabecular (porous, 'spongy' bone typically found inside bone organs) and cortical (compact, 'dense' bone, usually constituting the 'shell' of a bone organ). The large amount of bone surface relative to bone volume in trabecular bone compared to cortical bone may make it particularly sensitive to shifts in the bone (re)modelling balance [5]. Beyond the loss of bone volume fraction in the trabecular bone compartment, changes in tissue morphology may contribute to the deterioration of bone quality of osteoporotic patients. Because such osteoporosis-related changes to the trabecular bone micro-architecture form a link between the bone (re)modelling balance at a tissue level and the mechanical performance of the bone organ, they are key to our understanding of the disease.

Prominent among parameters considered when evaluating tissue-level morphological changes is structure model index (SMI) [6]. SMI was designed to estimate how far a trabecular geometry may be considered rod- or plate-like [7]. Evaluation of SMI across a number of datasets from human patients and animal models suggests that trabecular geometry transitions from being more plate-like to more rod-like as osteoporosis severity increases (plate-to-rod transition) [8–12]. However, it is well known that SMI correlates strongly with bone volume fraction, rendering dubious the comparison of SMI values between samples of considerably different bone volume fraction, such as osteoporotic samples versus healthy control samples. Furthermore, the concept of SMI is based on relative changes in surface area in response to a small dilation (a parallel offset from the surface), and relies on the fact that dilating a shape with non-negative Gaussian curvature (such as a sphere (SMI = 4), a cylinder (SMI = 3) or an infinite plane (SMI = 1)) never decreases the surface area. This is not the case in trabecular bone, because large parts of the trabecular bone surface are hyperbolic (saddle-like) [13], which causes local shrinking of the surface when the shape is dilated [14].

Ellipsoid factor (EF) has been proposed as an alternative method to measure the plate-to-rod transition in trabecular bone [15]. EF has since been used within and beyond bone biology, for example in bone surgical implant testing [16] and the characterization of the trabecular bone phenotype of genetic dwarfism [17], of the primate mandible [18], of the human tibia [19], of animal models of osteoarthritis [20], and fuel cell performance [21,22]. Apart from the original critique of SMI [14], as far as we know, there have been no further reports of EF in osteoporotic samples in the literature.

In this study, we expand on our two previous studies on the use of EF and the putative plate-to-rod transition in osteoporosis [14,15]. Specifically, we present new EF data on trabecular bone from an animal model of disuse osteoporosis (loss of bone mass as a consequence of reduced or altogether removed loading of the bone) as well as from human second lumbar (L2) vertebral bodies from women of varying age and bone volume fraction. The aim of the study is to investigate the association between variables describing the trabecular architecture (EF and SMI) and bone health. Our EF data relies on an updated and validated implementation of EF. This implementation is written for IMAGEJ2, and entails an additional way to seed new ellipsoids, and the ability to average EF over several runs. We also report the effect of input parameters more in-depth (details in the electronic supplementary material, (a)). EF is available freely as part of the latest BONEJ, a collection of IMAGEJ plug-ins intended for skeletal biology [23,24].

# 2. Methods

## 2.1. Ellipsoid factor algorithm

The EF algorithm was first reported in a previous study [15] and is explained here again owing to its fundamental relevance to the present study.

EF is a scalar value assigned to each foreground pixel in the three-dimensional binary image stack of interest. The EF of each pixel depends on the maximal ellipsoid that contains the pixel and that is contained in the image foreground. Denoting the semi-axis lengths of the maximal ellipsoid as $a$, $b$ and $c$ (with $a \leq b \leq c$), EF of each pixel is calculated as a difference of sorted semi-axis ratios:

$$\mathrm{EF} = \frac{a}{b} - \frac{b}{c}.$$

EF is confined between −1 and 1, with −1 being very plate-like, and 1 very rod-like (figure 1).

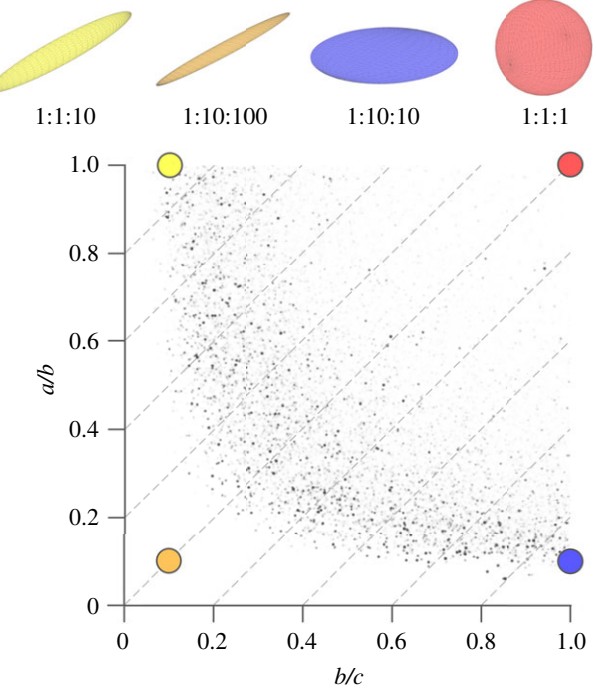

**Figure 1.** Ellipsoid factor (EF) is calculated as the difference of semi-axis ratios $EF = a/b - b/c$, where $a \leq b \leq c$ are the semi axis lengths. This figure shows edge cases of possible maximal ellipsoids (top row from left: 'javelin' (yellow), 'surfboard', (orange), 'discus' (blue), 'tennis ball' (red)), their semi-axis ratios (as $a{:}b{:}c$), and where a pixel (large, colourful points) within such an ellipsoid would be registered on the Flinn peak plot (the Flinn peak plot shows the semi-axis ratios of each maximal ellipsoid plotted against each other). Note that the orange and the red ellipsoid have the same EF, but vastly different Flinn peak point locations. Small black points are the Flinn peak plot data from the trabecular bone of a great spotted kiwi (*Apteryx haastii*) [25]. Ellipsoids with the same EF value, i.e. EF isolines, are represented by the grey, dashed diagonal lines with slope 1 on the Flinn plot. Flinn peak plots of trabecular bone typically exhibit a 'boomerang' shape.

EF is calculated by fitting locally maximal ellipsoids into the image foreground, then iterating over the foreground pixels to find the largest ellipsoid in which each pixel is contained. Note that the locally maximal ellipsoid is generally non-unique (electronic supplementary material, (ii)).

## 2.1.1. Ellipsoid fitting

First, points where a small sphere can start to grow (seed points) are determined. Two strategies for finding seed points are provided. The first is called distance-ridge based seeding. It involves subtracting the results of morphological opening and a closing operations on the distance transform of the input image from each other. The second is a topology-preserving skeletonization [26]. Calculating distance-ridge based seeds is computationally more efficient than skeletonization in practice, but may overestimate the number of seeds needed to fit a particular region and may miss thin features that skeletonization preserves well (figure 2).

After being seeded, each spherical ellipsoid grows uniformly by one user-defined increment at a time until a number of surface points equal to the user-defined 'contact sensitivity' parameter hit the trabecular bone boundary (a background pixel). Surface points are chosen from a random uniform distribution on the ellipsoid surface.

When the growing ellipsoid hits the trabecular bone boundary for the first time, the vector from the ellipsoid centre to the average contact point is set as the first ellipsoid semi-axis and the ellipsoid is contracted slightly. Growth of the ellipsoid then continues in the plane orthogonal to this first semi-axis, again until the boundary is hit. This initial ellipsoid fitting is followed by a series of small random rotations, translations and dilations of the ellipsoid in an attempt to find a larger ellipsoid in the local region. Further growth directions at this point are recalculated at each iteration to be random, but in the plane defined by the mean contact vector. These attempts end if no increase in volume of the ellipsoid is found after a user set maximum number of iterations (default 50, see the electronic supplementary material, (b)), or if the total number of attempts exceeds 10 times the

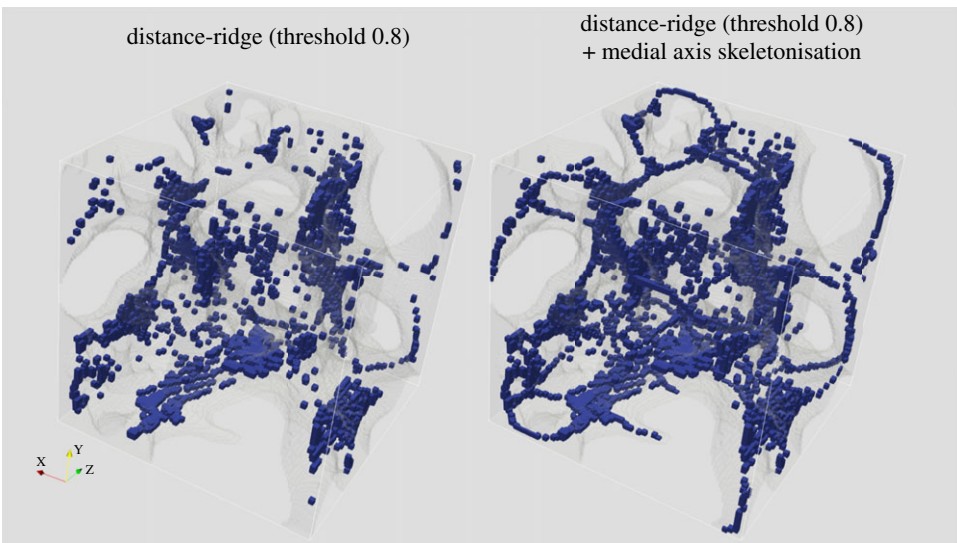

distance-ridge (threshold 0.8)

distance-ridge (threshold 0.8)
+ medial axis skeletonisation

**Figure 2.** The trabecular surface (transparent grey) from a lesser dwarf shrew (*Suncus varilla*) femur (a test image from a previous ellipsoid factor study [15]) and seed points (blue) from the distance ridge (left) and from the distance ridge and the skeletonization (right). The skeletonization seems likely to fit thin regions better, but the distance ridge allows more ellipsoids to seed in thick regions, which can lead to higher filling percentages overall, but may also seed more ellipsoids than necessary.

maximum iteration number. If more than half of the sampling points on the ellipsoid are outside the image boundary it is invalid, removed and ignored in further calculations.

### 2.1.2. Assign ellipsoid factor to each pixel and averaging over runs

Once maximal ellipsoids are found for each seed point, each foreground pixel is assigned the EF value of the largest ellipsoid that contains it, or NaN (not a number) if no ellipsoids contain that pixel. One iteration of fitting ellipsoids and assigning EF to each pixel is termed a *run*.

EF is a stochastic process and therefore results can vary from run to run. The user has the option to average the outputs over several runs to smooth the results. To track how this average evolves across runs, EF reports the 'filling percentage' (percentage of trabecular volume pixels that are contained by at least one ellipsoid) as well as the maximum and median EF change. The latter are estimates of how pixel EF values changed by adding the latest run. From experience on various real-life examples, we recommend averaging over six runs (the 'repetitions' input parameter) for the final result generation. This typically reduces the median and maximum EF change per pixel per run to less than 0.15 and 0.4, respectively (see the electronic supplementary material, (c)).

### 2.1.3. Ellipsoid factor inputs and outputs

Some further mathematical considerations on the shape of the distributions to be expected when calculating a difference of semi-axis ratios can be found in the electronic supplementary material, (d).

An overview of EF parameters (with default values) can be found in table 1, while EF results table outputs and output images are summarized in tables 2 and 3 respectively. In the present study, we ran EF on two datasets, with sample descriptions and statistical analysis detailed in the next two subsections. EF input parameters used for each of these studies are listed in table 4. For both studies, we measured bone volume fraction (BV/TV) and SMI, calculated descriptive statistics of the EF distribution (median, maximum and minimum), and plotted EF histograms.

## 2.2. Disuse osteoporosis in mouse tibiae

X-ray microtomography (XMT) scans (5 µm nominal pixel spacing) of 12 murine tibiae were obtained from an unrelated study [27]. The animals had undergone sciatic neurectomy to the right hindlimb, inducing one-sided disuse osteoporosis. They were divided into three groups of four mice. Groups 1, 2 and 3 were euthanised 5, 35 or 65 days after surgery, respectively. Trabecular bone from the proximal metaphysis was segmented by drawing around the trabecular-cortical boundary using the software CTAN (Bruker, Belgium).

**Table 1.** List of EF input parameter names, brief descriptions and default values, as listed in the ellipsoid factor documentation. (We suggest that users record and publish their values for these parameters as well as the BoneJ version used to enhance the reproducibility of their experiment (see table 4 for some examples). Note that although the defaults for 'skeleton points/ellipsoid' and 'repetitions' are 10 and 1, respectively, these values represent 'good' values to get a quick overview on an example image. Once ready to run on an entire dataset, we recommend setting these to 1 and 6, respectively (see the electronic supplementary material, (c)).)

| parameter name | description | default values |
|---|---|---|
| number of sampling vectors | number of sampling directions used to search for contacts with the boundary | 100 |
| sampling increment | increment for vector searching in pixel units | 1/2.3 |
| skeleton points per ellipsoid | number of skeleton points per ellipsoid. Sets the granularity of the ellipsoid fields | 10 |
| contact sensitivity | number of sampling vectors in contact with surface required to be classified as a collision | 1 |
| maximum iterations | maximum fitting iterations to try improving ellipsoid fit before stopping | 50 |
| maximum drift | maximum distance ellipsoid may drift from seed point. Defaults to unit pixel diagonal length | 1 |
| repetitions | number of separate runs over which to average EF value | 1 |
| seed points (distance ridge) | seed ellipsoids based on the foreground distance ridge | yes |
| seed points (topology-preserving) | seed ellipsoids based on topology-preserving skeletonization | no |
| show secondary images | display secondary images (volume, semi-axes, semi-axis ratios, Flinn plot) | no |
| show convergence data | display convergence data for two runs or more | no |

**Table 2.** Values written by EF into the ImageJ results table. (Median change $n$ and maximum change $n$ values are only shown if the 'show convergence data' input box is ticked and EF is averaged over at least 2 runs.)

| | |
|---|---|
| min EF | minimum of sample EF distribution |
| max EF | maximum of sample EF distribution |
| median EF | median of sample EF distribution |
| filling percentage | percentage of foreground filled by at least one valid ellipsoid |
| number of ellipsoids found | total number of valid ellipsoids fitted into trabecular foreground |
| median change $n$ | median change in EF value from run $(n\text{-}1)$ to run $(n)$. This indicates how well the EF algorithm converged |
| maximum change $n$ | maximum change in EF value from run $(n\text{-}1)$ to run $(n)$. This indicates how well the EF algorithm converged |

The segmented images were denoised using a three-dimensional median filter (radius 3 pixels) and thresholded at a pixel value of 75 (figure 3). The thresholding value was selected visually as sensible on one sample and kept consistent across samples. As the EF distributions were uni-modal and not normal in all cases, the EF median, maximum and minimum were taken as representative values for each specimen. SMI values were computed for each sample (using Hildebrand and Rüegsegger's method [7] with volume resampling 2 and mesh smoothing 0.5) using LEGACY BONEJ 1.4.3 [28] (https://github.com/bonej-org/bonej—SMI has been discontinued by MODERN BONEJ), and bone volume fraction measurements (calculated using CTAN) were taken from the raw data of an unrelated study [27].

For each group, paired $t$-tests comparing EF descriptive statistics (median, maximum and minimum), SMI and bone volume fraction between control and disuse leg were performed using the R software [29].

**Table 3.** EF primary (first four lines) and secondary (remaining lines) output images, with brief descriptions.

| | |
|---|---|
| EF image | image containing EF values for foreground pixels |
| seed image | binary image with ellipsoid seed points in foreground |
| volume image | image containing the volume of the locally maximal ellipsoid |
| ID image | image containing the index in sorted ellipsoid list |
| *a* image | shortest semi-axis of locally maximal ellipsoids |
| *b* image | intermediate semi-axis of locally maximal ellipsoids |
| *c* image | longest semi-axis of locally maximal ellipsoids |
| *a/b* image | *a/b* semi-axis ratio image |
| *b/c* image | *b/c* semi-axis ratio image |
| Flinn peak plot | plot of semi-axis ratios of locally maximal ellipsoids (*y*-axis: *a/b*, *x*-axis: *b/c*) |
| Flinn plot | plot of semi-axis ratios of all (not necessarily maximal) ellipsoids fitted |

**Table 4.** EF input parameters used for the two case studies presented in this article.

| study subject | mouse tibiae | human vertebrae |
|---|---|---|
| description in methods | (b) | (c) |
| number of vectors | 100 | 100 |
| sampling increment | 1/2.3 | 0.1/2.3 |
| seed points per ellipsoid | 1 | 1 |
| contact sensitivity | 1 | 5 |
| maximum iterations | 50 | 50 |
| maximum drift | 1 | 1 |
| number of runs | 6 | 6 |
| average of largest *n* | 1 | 1 |
| seed points (distance ridge) | yes | yes |
| seed points (topo.-preserv.) | yes | yes |

We also performed Pearson's product-moment correlation tests for association between EF median and bone volume fraction, and between SMI and bone volume fraction across all groups (using R's `cor.test()` function). The R scripts used for this purpose can be found in an online repository [30] under `/R/paired-mouse-disuse-test.R`.

## 2.3. Ellipsoid factor in human vertebrae of varying trabecular bone volume fraction

To investigate the association of SMI and EF with human bone health, we imaged sagittal sections of 22 vertebrae from women of varying age (24–88 years old) using XMT (30 μm pixel spacing). Pixels with a linear attenuation coefficient of more than $0.7\,\mathrm{cm}^{-1}$ were classified as bone, others as background. Cuboidal regions of interest containing trabecular bone, aligned with the image axes, were chosen manually. The vertebrae were originally collected and prepared for imaging with scanning electron microscopy in a previous study [31]. This dataset was interesting to the present study for two reasons. Firstly, these are the first EF numbers obtained on healthy and osteoporotic samples from humans. Secondly, they constitute a challenge for choosing reasonable EF input parameters because they are close to the resolution limit at which we can expect EF to fit the local shape well (trabecular thickness is approx. 5–8 pixels in these images). We additionally report mean and maximum trabecular thickness (Tb.Th (mm)) in these samples, in order to be able to gauge whether a change in EF might be attributed to badly resolved trabeculae in samples with e.g. low bone volume fraction.

The age distribution of our vertebral samples was non-normal, as it was skewed to the left by the prevalence of older samples (Shapiro-Wilk test $p < 0.05$). We therefore performed a non-parametric test of

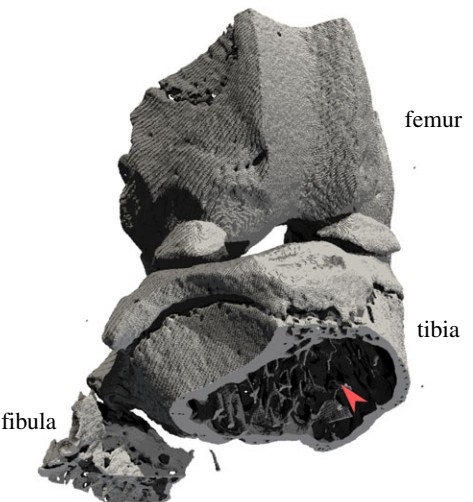

**Figure 3.** The right knee of one of the mouse samples rendered from a binary image. Red arrowhead points to the trabecular bone in the region of interest for this study. View is cranio-caudal with an oblique tilt towards proximal.

association of age with bone volume fraction. All other variables of interest (bone volume fraction, EF median, EF maximum, EF mode, EF minimum, SMI, SMI+, SMI-, mean Tb. Th., maximum Tb. Th) could be assumed to follow a normal distribution (Shapiro–Wilk test $p > 0.05$). As a consequence, we used Pearson's $r^2$ as a measure of association between these variables and bone volume fraction in our statistical correlation tests. All statistical analysis of the vertebral samples was based on a custom script (available at [30] under `R/histo-EF-stats-vertebrae-final.R`) using the R programming language [29].

# 3. Results

All distributions of EF observed in images of bone were uni-modal, as seen in the histograms of figures 4 and 13. As described earlier, we used the median, maximum and minimum (and the mode, for the vertebrae) of the distribution as a representative value to describe the distributions of local shape in these images for statistical analysis.

## 3.1. Disuse osteoporosis in mouse tibiae

EF images and histograms for our murine samples can be seen in figures 5 and 4, respectively. Paired one-sided $t$-tests ($n=4$) showed BV/TV and SMI values were significantly different between disuse and control limbs at the 5% level between control and disuse groups at all time points (figures 6 and 7). Minimum and maximum EF were not statistically associated with disuse ($p > 0.05$) at any time point (figure 8). There was no link between EF median and disuse at 5 days (paired one-sided $t$-test, $p > 0.05$) and 35 days (difference not normally distributed (Shapiro–Wilk $p < 0.05$), paired one-sided Wilcoxon rank sum test, $p = 0.06$), but there was a statistical difference at 65 days ($p < 0.05$). Unlike SMI, these measurements suggest therefore the presence of a small shift of about EF 0.1 occurred only after a large amount of bone had already been lost. Over all time points, there was a considerably less strong, and less statistically significant, relationship between bone volume fraction and EF median (Pearson's $r^2 = 0.25$, $p < 0.05$) than between bone volume fraction and SMI (Pearson's $r^2 = 0.81$, $p < 0.001$, figure 9). The R-script used to perform this analysis can be found under `/R/mouse-smi-tests.R` in [30]. EF filling percentage was higher than 90% for all our murine samples, although significantly differed between disuse and control at all time points (paired $t$-test, $p < 0.05$).

## 3.2. Ellipsoid factor in human vertebrae of varying trabecular bone volume fraction

Filling percentages ranged from 74% to 97% and median change in EF between the two final runs ranged from 0.1 to 0.17 (figure 10), which suggests EF reached a reasonable level of convergence at this point. Correlation tests showed that there was no association ($p > 0.05$) between bone volume fraction and any of the three convergence variables; median change, maximum change and filling percentage, indicating that the EF algorithm did not preferentially fill the trabecular bone more completely or in a

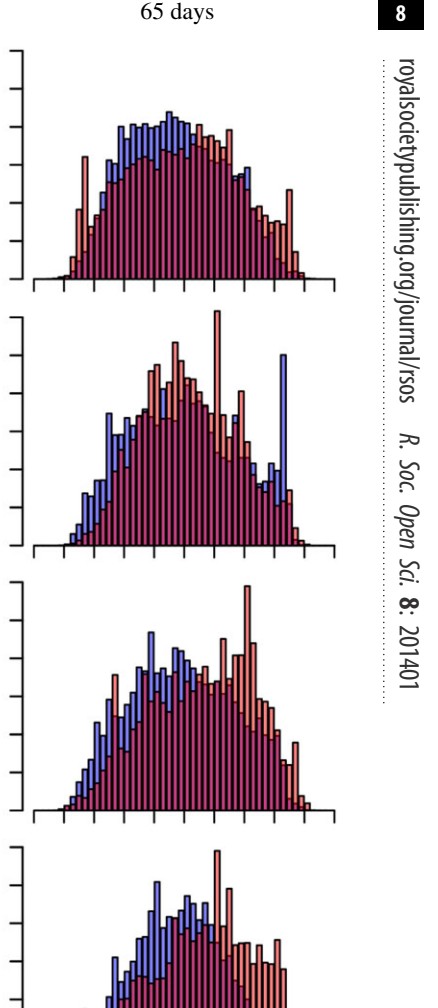

**Figure 4.** EF frequency histograms for each mouse, at 5 (left column), 35 (central column) and 65 (right column) days post-surgery. Large parts of the control (blue) and disuse (red) histograms overlap (purple). Paired *t*-tests on EF median suggest a subtle plate-to-rod-transition at 35 and 65 days, but no plate-to-rod transition despite significant bone loss at 5 days.

more stable way in samples with relatively low or high bone volume fraction. This was evidence for a satisfactory convergence of the EF algorithm, albeit not as complete as in the murine samples.

There was a negative association between BV/TV and age (Spearman's $\rho = -0.58$, $p = 0.004$), but not between BV/TV and mean or maximum trabecular thickness ($p > 0.05$). The latter result suggests that EF measurements are unlikely to be affected differentially by the resolution of the trabeculae in samples with lower bone volume fraction. It might also imply that lower bone volume fraction is not associated with thinner 'trabeculae' (but rather 'fewer'), although this was not a focus of the present study.

SMI, SMI+ and SMI- were strongly, negatively and significantly associated with bone volume fraction: Values for Pearson's $r^2$ were 0.47, 0.42 and 0.53, respectively, while *p*-values were all <0.005 (figure 11). SMI ranged from 1.36 to 3.11.

Median, maximum and minimum EF were not associated with bone volume fraction ($p > 0.05$; figure 12), and there was a mild negative association between bone volume fraction and EF modal value ($r^2 = 0.2$, $p = 0.03$). Histograms of the EF distribution were occasionally skewed in either direction across all values for bone volume fraction (figure 13). Sometimes similar EF values clustered in one region of the vertebra,

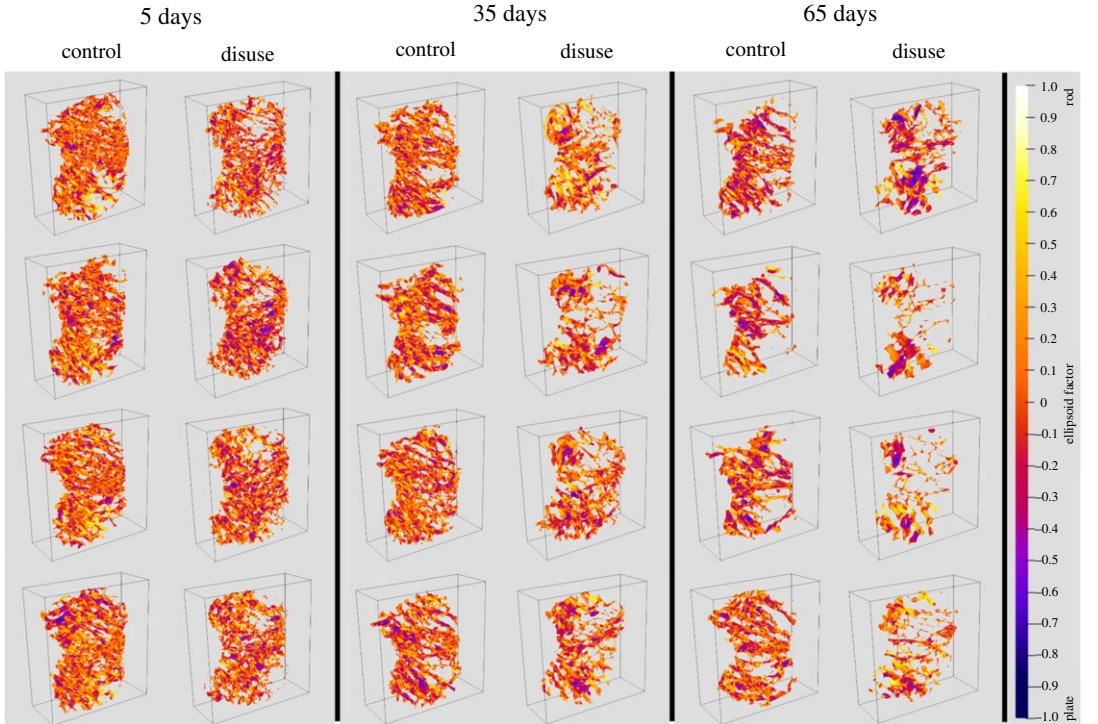

**Figure 5.** EF renders for each mouse limb over the three time points. The positive $Y$-direction is towards posterior and the positive $Z$-direction is towards proximal. The $X$-direction is towards lateral for the control limbs, and towards medial for the disuse limbs. Bone loss appears to start medially.

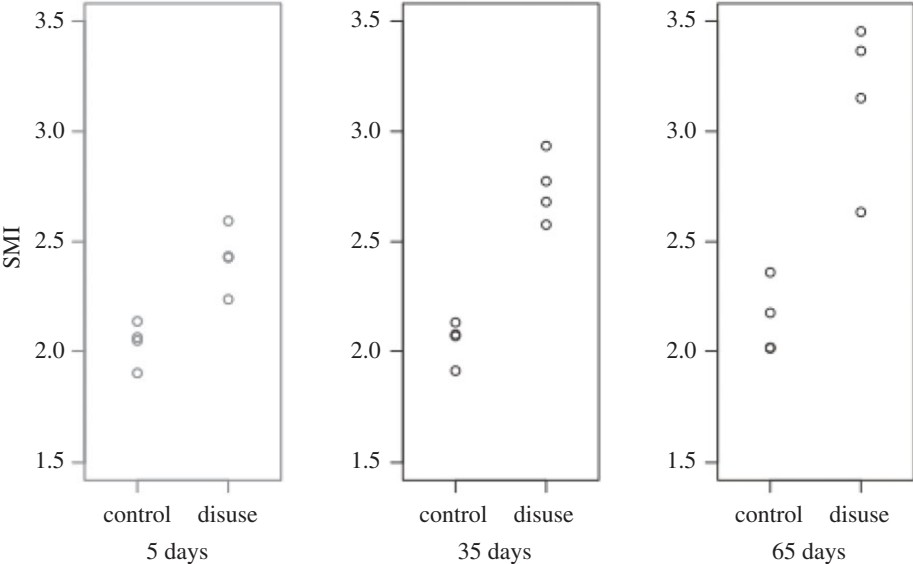

**Figure 6.** SMI at each time point for control and disuse mouse limbs. SMI is significantly different between control and disuse limb at all time points ($p < 0.05$).

while in other cases, a range of EF values could be found in all anatomical regions considered. Figure 14 shows EF images for 20 of the 22 vertebrae we analysed.

## 4. Discussion

We measured EF distributions in trabecular bone from healthy and unloaded mouse tibiae and from human vertebrae. Only on some occasions, EF supported the presence of a small shift towards a more rod-like geometry linked with decreases in bone volume fraction. SMI, on the other hand, suggested

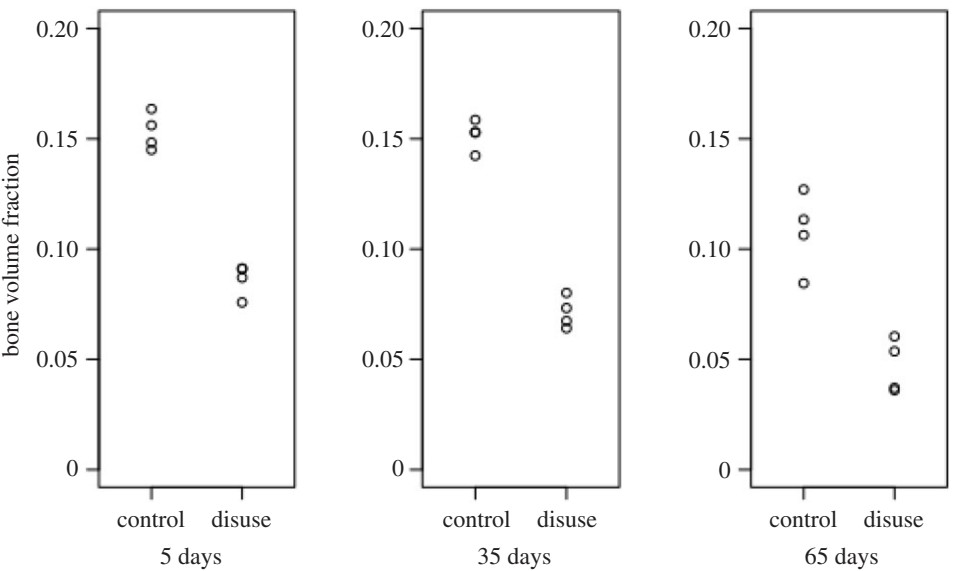

**Figure 7.** Bone volume fraction at each time point for control and disuse mouse limbs. Bone volume fraction is significantly different between control and disuse limb at all time points ($p < 0.05$).

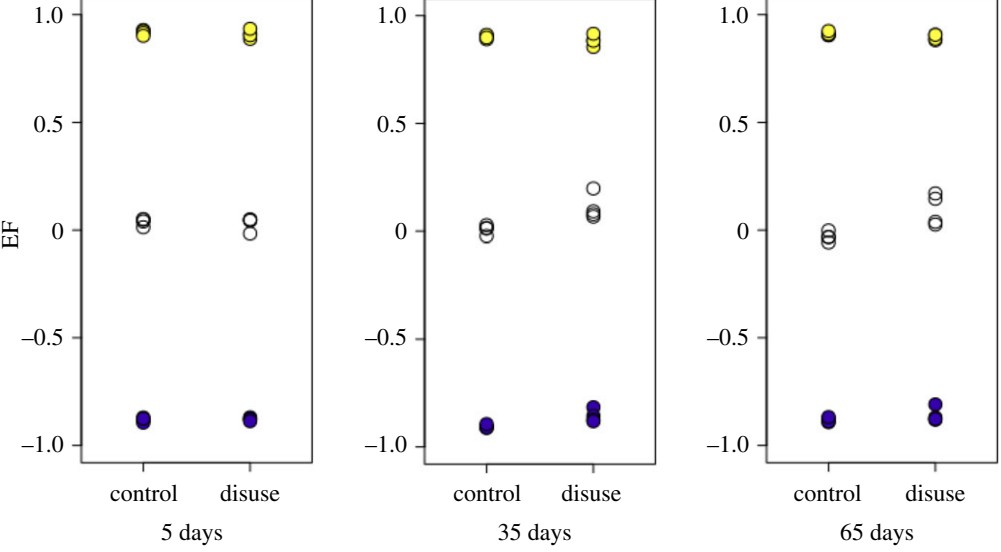

**Figure 8.** EF median (white), maximum (yellow) and minimum (blue) at each time point for control and disuse mouse limbs. EF median is significantly different ($p < 0.05$) between disuse and control limbs only at 65 days.

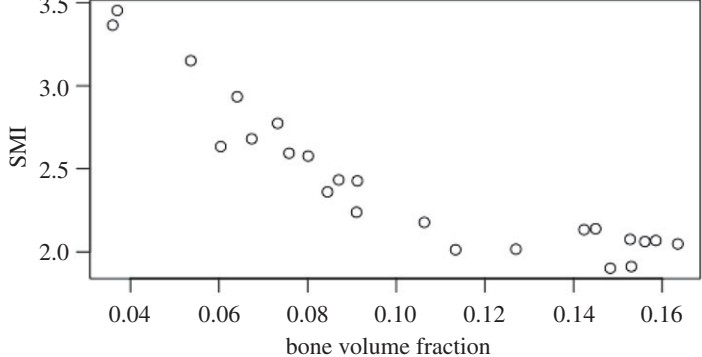

**Figure 9.** SMI value plotted against bone volume fraction in murine trabecular bone samples. Bone volume fraction and SMI values are strongly, negatively correlated (Pearson's squared moment-product correlation $r^2 = 0.81$, $p < 0.001$).

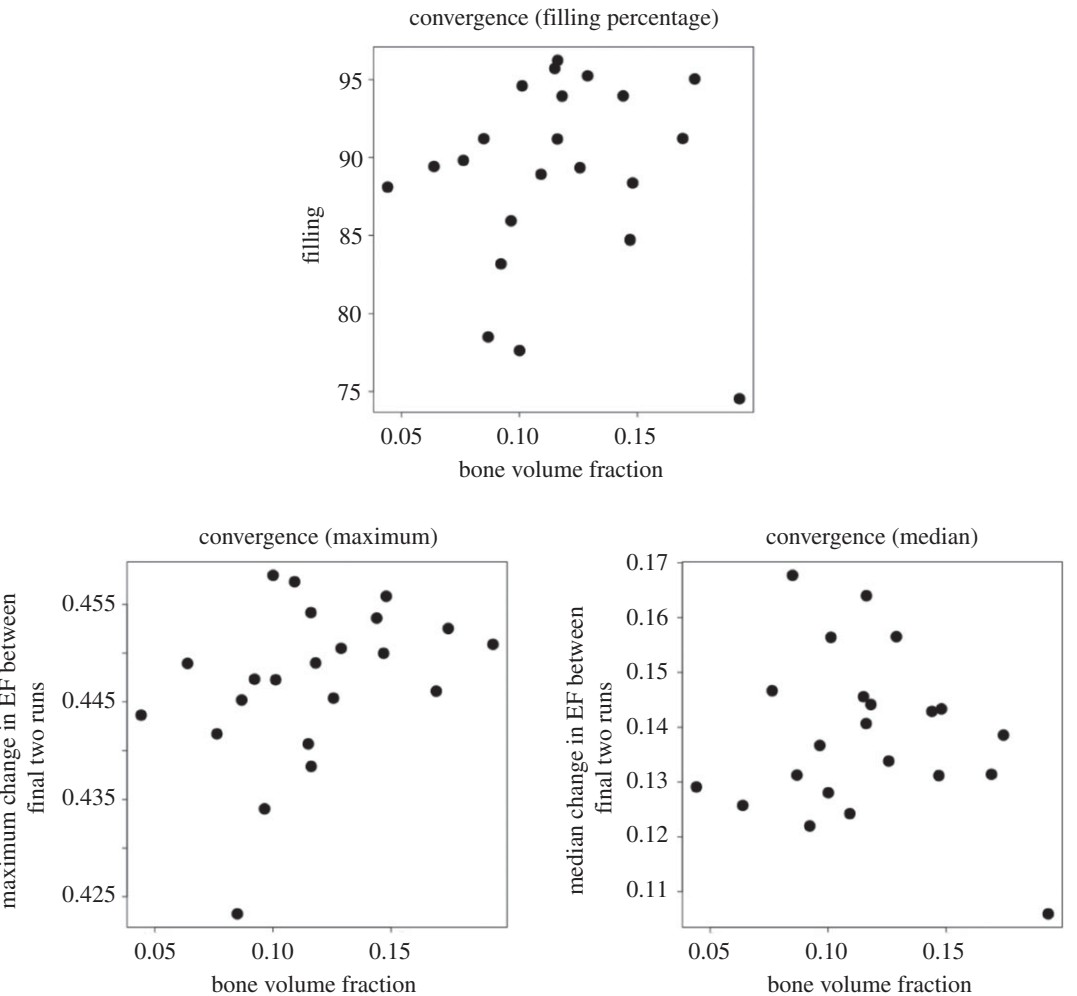

**Figure 10.** EF convergence parameters (filling percentage, maximum and median EF change between two final runs) plotted against bone volume fraction for our human vertebral samples. There were no statistical associations between the variables, showing that all samples were equally likely to have a high filling percentage, independent of volume fraction.

the presence of a drastic plate to rod transition whenever a difference in bone volume fraction was found. EF distributions in the samples from both species we investigated in the present study were consistently uni-modal.

In the murine samples, bone loss happened shortly after surgery in one condyle, but EF median changed only later during disease progression. This suggests that local shape changes in the trabecular bone may be delayed with respect to the initial loss of bone. The strong interdependence between SMI and bone volume fraction is misleading in this case, as it supports an immediate change in local trabecular shape that implies a geometry that is more rod-like than a perfect rod (SMI > 3, tending towards spherical where SMI = 4) at the latest time point. EF median was mildly and only just significantly associated with bone volume fraction across all samples. We interpret this as a possible subtle tendency towards a more rod-shaped local geometry in some samples, which would be impossible to glean from observing SMI alone. Minimum and maximum EF values are not different in healthy and osteoporotic murine samples, underlining that very plate- and very rod-like structures coexist in all samples.

Similarly, in the human vertebra samples, only the mode of the distribution correlated with bone volume fraction, highlighting that any changes in local shape linked to a decrease in bone volume fraction are subtle. Considerable variability in local shape can be seen in the EF images of the vertebral samples. Some of the samples agree with the results of a descriptive anatomical study of human fourth lumbar vertebral bodies, which characterized the trabecular geometry as central plates and braces surrounded superiorly and inferiorly by a honeycomb of rods [32].

In this study, we further presented some recommendations for suitable default parameters for EF (table 1), based on the convergence behaviour of EF reported in the electronic supplementary material, as well as an overview of the EF results (table 2) and output images (table 3).

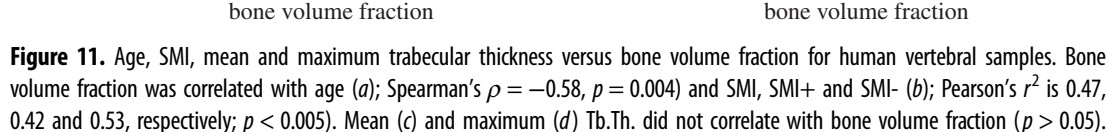

**Figure 11.** Age, SMI, mean and maximum trabecular thickness versus bone volume fraction for human vertebral samples. Bone volume fraction was correlated with age (*a*); Spearman's $\rho = -0.58$, $p = 0.004$) and SMI, SMI+ and SMI- (*b*); Pearson's $r^2$ is 0.47, 0.42 and 0.53, respectively; $p < 0.005$). Mean (*c*) and maximum (*d*) Tb.Th. did not correlate with bone volume fraction ($p > 0.05$).

## 4.1. What is the mechanical relevance of plates and rods in cancellous bone?

Considering trabecular bone a 'cellular solid' [33] gave rise to the idea that plates and rods contribute to mechanical performance. Theoretical, idealized models of open-cell and closed-cell porous solids predicted a dependence of the stiffness and strength on the square and the cube of the characteristic length *r*, respectively. In a seminal study for the concept of rods and plates in trabecular bone, Gibson analysed previous data from this perspective and showed that these models were consistent with a transition from open-cell to closed-cell mechanical behaviour at a bone volume fraction of 0.2 [34]. This is further evidence that attempting to measure rods and plates in trabecular bone is not independent of the amount of bone present (contrary to what was stated in the original SMI study [7]). The bone volume fraction in our samples was below 20%, where the influence of hyperbolic parts of the surface and negative SMI are less than in samples with greater BV/TV [14], so it would be interesting to compare EF in samples with bone volume fraction above and below this value in the future.

The mechanical environment has a strong effect on bone size and shape at an organ and tissue level (e.g. [35–37], for a review, see [38]), but Frost's mechanostat may not be the main driver of trabecular adaptation within the life of an individual [5]. Across species, trabecular bone micro-structure scales as a function of animal size and is likely to behave differently in small animals compared to large animals [39].

Changes in local shape may indicate preferential osteoclastic resorption and/or osteoblastic formation in certain areas of bone. Qualitative descriptions based on scanning electron micrographs of human lumbar vertebrae suggest defective and or slowed bone formation and mineralization, as well

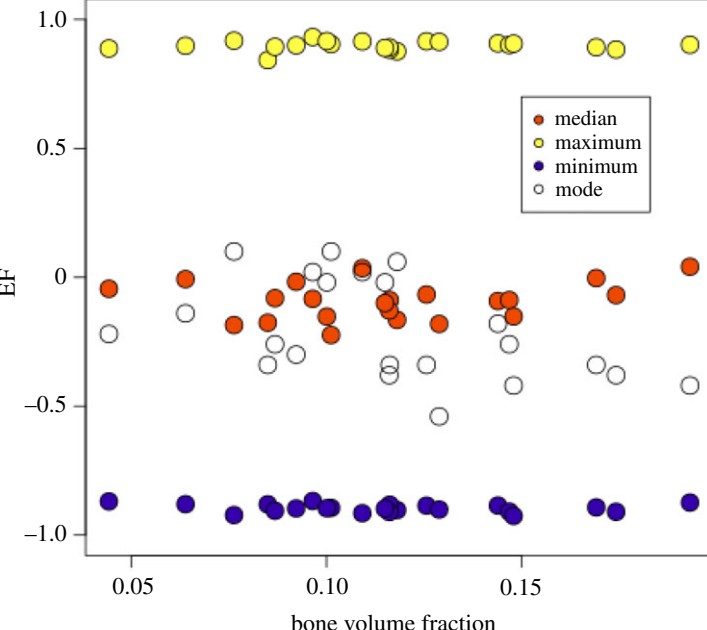

**Figure 12.** EF distribution parameters plotted against bone volume fraction in human vertebral samples. Only the mode of the distribution was mildly associated with bone volume fraction; median, maximum and minimum were not.

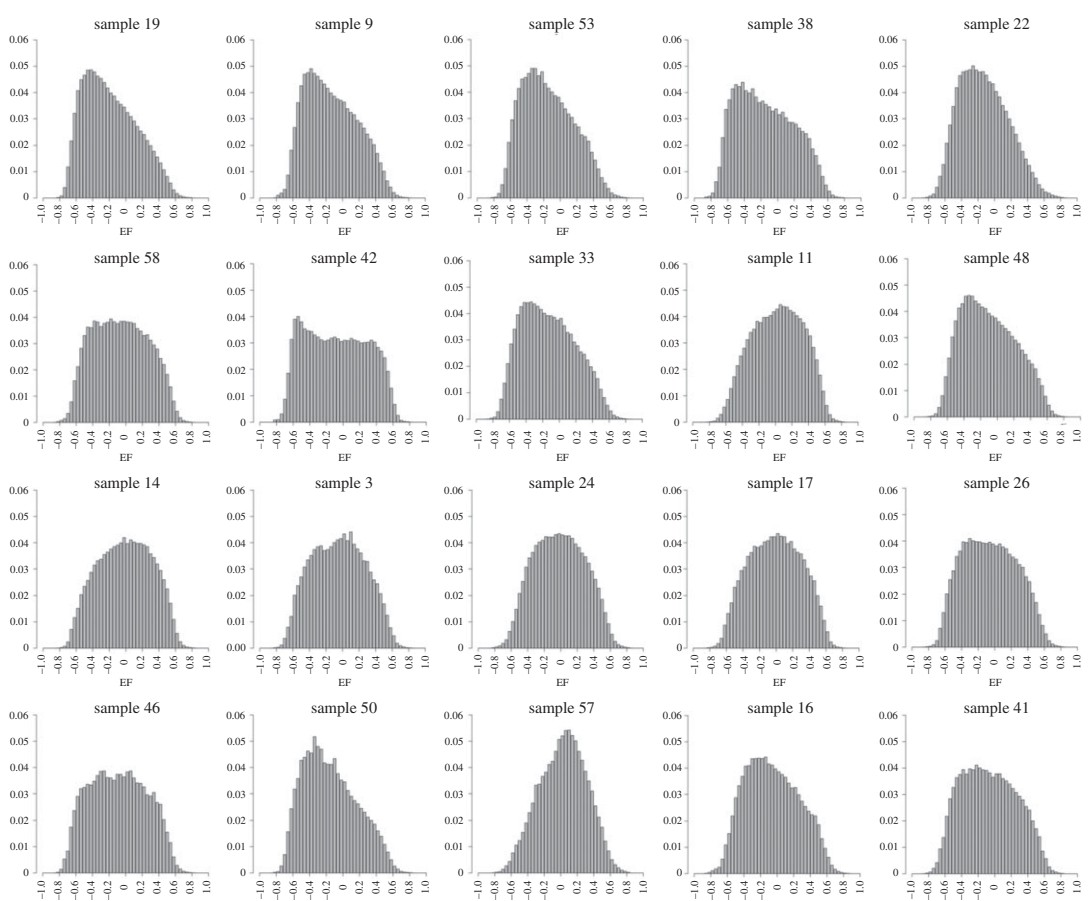

**Figure 13.** EF histograms of human vertebral trabecular bone samples, sorted from top-left to bottom-right by bone volume fraction. Images of the samples are shown in the same order in figure 14. EF histograms are all unimodal, and skew in either direction. The mode, but not the median, of the EF distribution was associated with bone volume fraction. Samples 49 and 12 are not shown owing to the lack of a reasonable way to arrange 22 figures on a page (there were several other samples with the same volume fraction), but were included in the analysis.

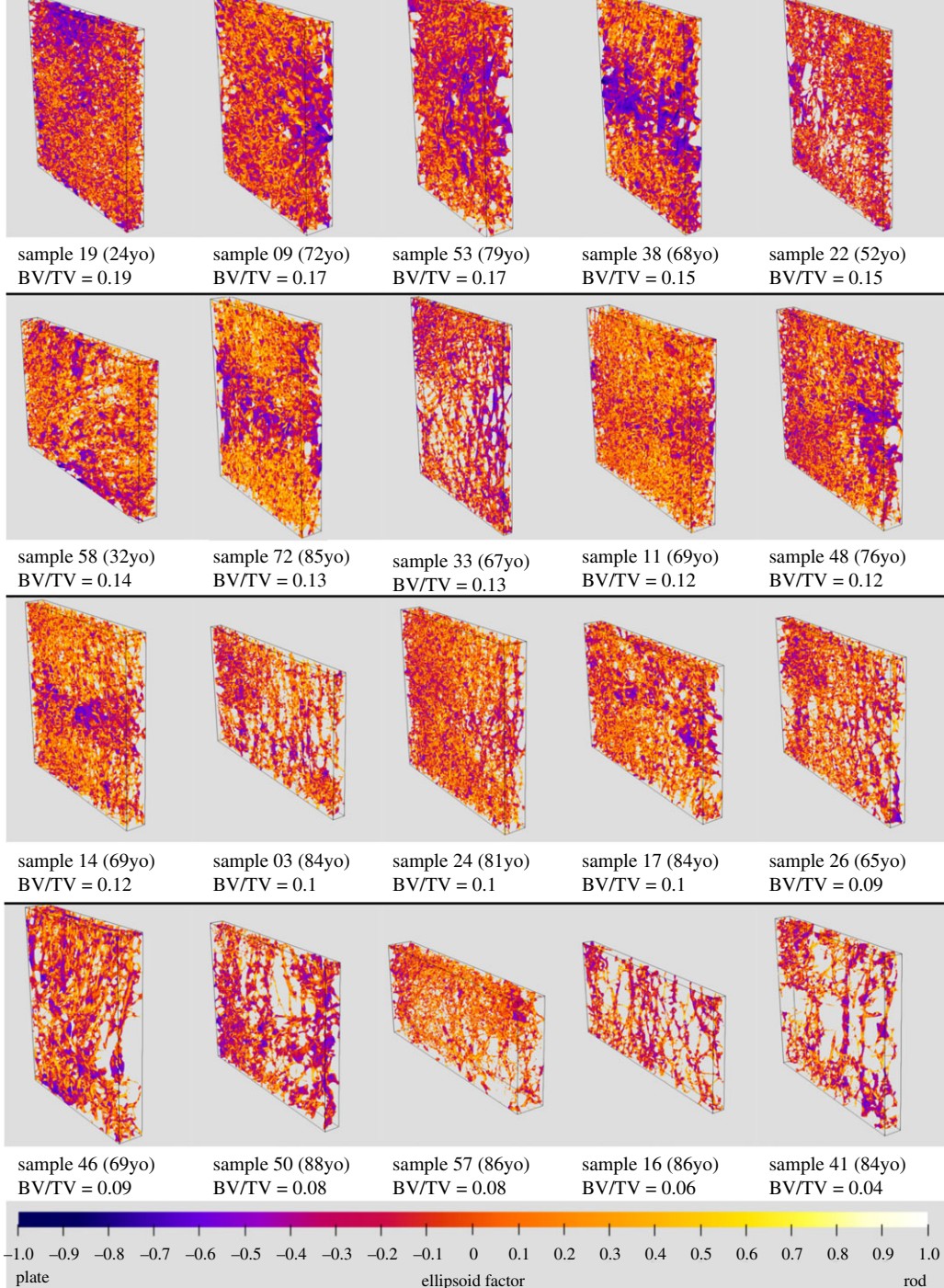

**Figure 14.** EF image renders of the vertebrae studied, sorted from top-left to bottom-right by bone volume fraction. Yellow pixels indicate a more rod-like, and blue pixels a more plate-like local shape, with orange indicating a shape on a continuum between plates and rods. Histograms of the EF values are shown in the same order in figure 13. All samples display a range of EF values. In some samples, pixels of similar EF value seem to cluster in the same region (e.g. sample 42), while in others there seems to be a mix of EF values in all regions (e.g. sample 33). Samples 49 and 12 are not shown owing to the lack of a reasonable way to arrange 22 figures on a page (there were several other samples with the same volume fraction), but were included in the analysis. Slices are parasagittal with superior towards the top of the images.

as decoupling of resorption and formation as characteristic of the osteoporotic trabecular geometry at a length scale below the one investigated in the present study [40]. Resorption cavities in human fourth lumbar vertebrae may occur most often near trabecular nodes, with the next most common location plate-like trabeculae [41]. The study gives no details on how plates, rods, nodes and

'fenestrations' are characterized. It would be interesting to correlate SMI and EF results with such observational studies in the future.

## 4.2. Measures of local shape beyond structure model index and ellipsoid factor

Individual trabecula segmentation (ITS) has been proposed as a method to classify the local shape of trabecular bone as rods and plates [42]. ITS is based on a decomposition of the trabecular geometry into surfaces and curves [43], with subsequent assignment of each foreground pixel to one of these surfaces and curves based on a measure of vicinity and orientation [44]. ITS has been measured in biopsies of hip replacement patients with inter-trochanteric fractures [45]. Compared to cadaveric controls, these fracture patients had lower ITS plate bone volume fraction, but equal ITS rod bone volume fraction, as well as lower stiffness moduli and lower overall bone volume fraction (BV/TV). We find it interesting that ITS-measured plate volume fraction correlates with stiffness in these studies. However, we note that ITS-measured axial volume fraction is also (often more strongly) correlated to stiffness than plate volume fraction. It is clear that, at equal bone volume fraction, bone that is less aligned to the direction in which stiffness is measured will behave in a more compliant manner than bone that is more strongly aligned to this direction [46]. We therefore suggest that, in the ITS studies, the driving factor for these observations may not be a change in local plate/rod shape, but rather a change in local alignment to the axes in which stiffness is measured. It would be interesting to compare ITS and EF results in the future.

Another method that decomposes trabecular bone into rods and plates was developed but validated only on objects with non-negative Gaussian curvature [47]. Applying it to human vertebral samples suggested that three parameters of micro-architecture (two relating to the supposed rod elements) explained 90% of bone stiffness, the same amount of variation in sample stiffness explained by apparent bone volume fraction alone [48]. However, all three of these parameters had a significant and strong correlation with bone volume fraction, and this study therefore does not constitute evidence for geometrical changes in the trabecular compartment driving mechanical properties beyond the loss of material. Fatigue failure of trabecular bone may further be related to elements oriented transversely to the main loading direction, which have little effect on stiffness and strength [49].

## 4.3. Limitations and future work

EF is a useful addition to the many geometrical and topological quantities that are routinely measured in trabecular bone, some of which depend on each other, as we have shown here. EF is at least designed to be a priori independent of bone volume fraction, the most important descriptor of trabecular bone mechanical properties [46,50]. The lengths of the ellipsoid semi-axes $a,b,c$ as half-thickness, half-width, and half-length trabecular variables could be seen as an extension to measuring trabecular thickness alone.

The fact that EF is generally non-unique (electronic supplementary material, (ii)) is at least in theory a limitation of EF. We doubt that ellipsoids found by EF in real-life trabecular structures will often be in this situation in practice.

The samples we consider in this paper are cross-sectional, which unfortunately precludes us from following the trabecular architecture of a single individual over time. EF, like all other measures of trabecular micro-architecture, requires a sufficient resolution of the individual geometrical features to minimize artefacts such as noise and partial volume effect. Where resolution is insufficient for EF to run on a binarized image, it might be possible to locate the trabecular boundary using fuzzy edge detection (and therefore circumventing the need for precise thresholding), as is done in the tensor scale algorithm [51–53]. The current EF software is designed in such a way as to make an approach based on fuzzy boundary detection straightforward. Very small trabeculae may be routinely missed by XMT altogether, but dealing with this limitation was outside the scope of this study.

EF is a complex algorithm, with several input parameters that need to be tailored to the application. We believe that this is also an advantage in some ways, as it will force users to better understand the methods they are using. We encourage users to ask questions can be asked on the IMAGEJ forum (https://forum.image.sc/). Despite its complexity, an advantage of EF is that it reduces local shape down to a single number per pixel. Important information on the subtlety of trabecular local shape is lost owing to this simplification and users are encouraged to view and interpret the Flinn peak plot because it is a more complete, but more complex, representation of the local shapes present in their sample (figure 1). The Flinn plot may require more advanced statistics, for two-dimensional, non-independent response variables, to rigorously compare sampled groups. It might be possible to

improve the performance of EF in the future by transferring some parallel computations onto the graphics card [54].

The analyses in the present study focus on median, maximum and median EF, and therefore treat the distribution of local shapes as a whole. This approach does not take into account the orientation of the ellipsoids, or whether pixels of a certain EF cluster around particular anatomical regions. Such considerations might be interesting to explore in the future, as would combinations of EF measurements with other measures of trabecular local shape.

Note that because EF was designed to be independent of scale (as it is a difference of ratios), it may assign the same EF to a 'small', mechanically unimportant and a 'large', load-bearing trabecula of the same local shape. Features are weighted by their size in the histogram and Flinn peak plot, because each pixel contained by an ellipsoid is assigned the EF of that ellipsoid. Consequently large and mechanically important features containing numerous pixels make a greater contribution to the final analysis, and small features may have a negligible effect. On the other hand, this intentional feature of EF may be useful to detect situations where a pure change in shape (without differences bone volume fraction or local thickness) is responsible for altered mechanical properties. Torres *et al.* [49] found that the size, shape and orientation of small off-axis members can be critical for the mechanical performance of porous materials. The maximal ellipsoids' EF, orientation and volume could provide this additional information.

Further avenues of future research could investigate how well EF characterizes curved trabecular bone, and understanding whether characteristic combinations of semi-axis ratios $a/b$ and $b/c$ for an individual or a group exist that are not immediately recognized by looking at the semi-axis ratio difference, but which may emerge in the Flinn peak plot. Extracting the ellipsoid semi-axis directions might also provide interesting information about the orientation of local shapes. For example, the local orientation may provide clues about the dominant loading environment in different regions of the trabecular compartment.

## 5. Conclusion

Our investigations suggest that local shape in trabecular bone is not straightforward to decompose into rods and plates, and that a wealth of shapes across the plate-rod continuum exist in any sample. Our data support the presence of a slight tendency of the trabecular geometry to have higher EF in osteoporotic samples, possibly as a consequence of a cell-driven re-organization that is delayed in respect to the initiation of bone loss. This transition, where it occurs, is considerably more subtle than SMI values suggest.

Ethics. The image data were obtained and re-used from unrelated experiments in which animal procedures and human samples were used with appropriate ethical approval. The use of animals in the unrelated study was carried out in accordance with the Animals (Scientific Procedures) Act 1986, an Act of Parliament of the UK, approved by the Royal Veterinary College Ethical Review Committee and the United Kingdom Government Home Office, and followed ARRIVE (Animal Research: Reporting of *In Vivo* Experiments) guidelines. Human second lumbar vertebral body samples were obtained via the European Union BIOMED I study 'Assessment of Bone Quality in Osteoporosis'.

Data accessibility. Vertebral XMT scans (https://figshare.com/projects/Assessment_of_Bone_Quality_in_Osteoporosis_-_XMT_and_SEM/76962) and segmented XMT of mouse trabecular bone (https://figshare.com/articles/dataset/segmentations_of_mouse_trabeculae/12200438) are available on the data sharing repository figshare. Some vertebral scans can additionally be explored as three-dimensional renderings on SketchFab: https://sketchfab.com/alexjcb/collections/vertebrae-sections. The BONEJ source code can be found on Github https://github.com/bonej-org/BoneJ2, with installation instructions at https://imagej.net/BoneJ2#Installation, and is archived within the Zenodo repository (https://doi.org/10.5281/zenodo.1427262). Custom Python, R and IJ Macro scripts [30] for the analyses presented here have also been archived within Zenodo (https://doi.org/10.5281/zenodo.3556577).

Authors' contributions. M.D. had the idea for ellipsoid factor and designed the overall project. A.A.F. and M.D. implemented the IMAGEJ code. A.A.F. wrote the custom R, IJ and Python helper scripts for the analysis [30], did the statistical analysis, made the figures and wrote the manuscript draft. R.D.S. performed the surgery and dissected the mouse tibiae. B.J. imaged the murine samples. S.M. segmented the trabecular compartment of the mouse tibiae, and measured bone volume fraction in the mouse samples under the guidance of B.J. D.M. and A.B. prepared and imaged the human vertebrae. All authors read the manuscript, provided feedback and approved the final version of the manuscript.

Competing interests. M.D. was a member of the Editorial Board of Royal Society Open Science at the time of submission and was not involved in the assessment of this submission.

Funding. This research was supported by a BBSRC Project grant to M.D. (BB/P006167/1).

**Acknowledgements.** The authors acknowledge the reviewers for their comments that helped improve the manuscript. The authors further thank Phil Salmon (Bruker Micro-CT), Andy Pitsillides (RVC) and the wider RVC Skeletal Biology Group for helpful discussions on trabecular bone, as well as Richard Domander (RVC) and Curtis Rueden (University of Wisconsin-Madison) for valuable help and support in working with Image J2. We also thank Yu-Mei Chang for advice with statistical analysis, and Eva Herbst for critically reading the manuscript.

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
