## [Peer Review File · Royal Society Open Science]

Review History

RSOS-201401.R0 (Original submission)

Review form: Reviewer 1

Is the manuscript scientifically sound in its present form?

Yes

Are the interpretations and conclusions justified by the results?

Yes

Is the language acceptable?

Yes

Do you have any ethical concerns with this paper?

No

Have you any concerns about statistical analyses in this paper?

No

Recommendation?

Accept with minor revision (please list in comments)

Comments to the Author(s)

This well-written manuscript presents a well-reasoned, reproducible set of analyses that show the disadvantages of the SMI measure and provides a clear implementation of the superior EF measure of trabecular geometry. Moreover, it adds to the growing body of evidence that the plate-to-rod transition, supposed to occur with trabecular bone loss, is subtle if it exists at all. I would recommend publishing this manuscript after the authors have addressed some minor revisions outlined below using the authors' line numbers. Chief among these in my opinion, are the extent of the non-uniqueness problem visualised in fig S9 and the fact that EF doesn't seem to be completely independent of BV/TV, the major criticism of SMI. I believe minimal addition to the discussion will remedy these problems.

Minor revisions:

L7 & 11: Please be consistent with the terminology and only use remodelling balance or (re)modelling balance.

L14-15: Consider rephrasing this sentence so that the verb is not at the end. E.g. "SMI was designed to estimate how far a trabecular geometry may be considered rod- or plate-like"

L19: Please replace 'vastly' with something to the effect of 'considerable' or 'notable'.

Interspecific differences in BV/TV are often absolutely larger than that measured between osteoporotic and murine controls here, so vastly is subjective as currently used.

L32-39: While I do not disagree with anything in this paragraph I would encourage the authors to highlight the updates made to the EF approach since Salmon et al., 2015. This expansion of earlier work is not simply an application to new osteoporotic data.

L43 & throughout: Here the ellipsoid vector is referred to as an axis, should this not be semi-axis?

L47, Supp. ii & Supp Fig. 9: The non-unique aspect of EF for certain voxels appears to be an issue for the method and should be briefly discussed in the limitations section. In the case presented in the supplementary material, which EF (-1 or 1) is reported for the voxels in the intersections of the two ellipsoids, the first found stochastically by the algorithm? Also, if several runs are performed will the intersecting voxels, with near -1 and near 1 EF values in different runs, be averaged to ~0? Lastly, if the maximal local ellipsoid is 'generally' non-unique then how much of a problem is this for the method? Presumably most trabecular structures will not have exactly the same volume and very different EF as in the extreme example of Fig. S9?

L50-79: This conceptual explanation is clear and well-written

L54-56: Please provide a reference(s) for the disadvantages of distance ridge seeding or briefly explain to the reader why these occur.

L63-64: Please clarify how the third semi-axis is found, presumably two vectors are found in the plane orthogonal to the first?

L64: Please replace 'following' with 'followed'

L71 & throughout: Since ellipsoids are 3D shapes should it not be 'voxel' instead of 'pixel'?

L72-79: I would encourage the authors to explicitly outline the three convergence variables here as otherwise the reader first encounters them, in-text, in the results section.

L86: Why was the median chosen here above mean or mode? The choice of average doesn't appear very important given the distributions depicted but given the mode of EF is significantly related to BV/TV why not report all 3 averages?

L95: Though not necessary, it would be good to know the kernel size of the median filter used for de-noising the images.

L100-101: Please clarify how BV/TV was measured, with BoneJ?

L118-119: In this section, please explain why Tb.Th is measured.

L123: Please change Tr.Th to the standard Tb.Th abbreviation.

L138-141: Please amend the statement concerning the 35 days results in figure 7 to match this reporting.

L143-144: Only correlation was mentioned in the methods section, yet regression appears to be used here. Please be consistent in reporting r or r^2 or explain why one was used

instead of the other. Please also report the effect size of the t-tests as the author's have done for the correlation/regression analyses.

L143-144: This result means that EF is related to BV/TV, though to a much lower extent than SMI. As this is a major, and justified, criticism of SMI this result should be addressed in the discussion.

L146: Figure 4 is only referenced in text after figure 8, please consider reordering these figures.

L147: This is the first mention of filling percentage, please explain what this is earlier in the manuscript.

L152: Context around the convergence parameters would help the reader understand why the authors are referring to the final 2 runs of 6 here.

L159: As above, please explain in the methods section why thickness is reported here. Does this implicitly demonstrate plates are not becoming significantly thinner with age?

L161: The r values could be easily confused with the r^2 values given in the previous section, please be consistent or more clearly signpost this for the reader.

L165: Please explain why mode is now used as well as the median? Or report all three common averages from the start. Also, the EF to BV/TV association, $r = -0.45$, is described as 'mild' but is still a significant relationship of the same order as -0.65 found for SMI. Perhaps it is better state it is 'lower' than for SMI.

L168: Why were 2 vertebrae not included in figure 13?

L182-184: This observation is very interesting.

L190: Please replace 'support' with 'supports'.

L191: Please replace 'culminates into' with 'culminates in'.

L197-198: Do the authors think local shape variability might also be seen in the murine sample if it were volumetrically larger, i.e. with a higher trabecular number?

L195: Modelling *in silico* and biological trabecular modelling could be confused here, I suggest changing this word. Similarly this is the first reference to 'cancellous' as opposed to 'trabecular' bone, I suggest changing to trabecular bone here for consistency.

L250: As above, why do the authors think that the EF measure was not totally independent of BV/TV, as it was designed to be? I completely agree with the authors that EF appears to be a superior measure of geometry relative to SMI, as it is less dependent on BV/TV, but this is still a result worthy of discussion.

Acknowledgements

L295: This section mentions Python scripts but I cannot find another mention of these in the manuscript. Please clarify or remove.

References

It was not clear initially to me how to access the R/python scripts from reference 27 – perhaps the URL or DOI would be more helpful? https://zenodo.org/record/3556577#.X_NGIfn7SUK

Figures

Figure 1 is intuitive and nicely demonstrates the shape continuum which is key strength of the EF approach. Do all samples conform to this distribution when expressed as a Flinn peak plot, and if so does this tell us anything about the most common shapes found in trabecular bone?

Figure 3. Perhaps add '(purple)' after 'overlap' in the caption

Figure 4. Consider reordering this figure as per above comment. Please also, give an origin and axes for the axis to aid the reader.

Figure 7. See results reporting comment above relating to 35 days.

Figures 8 & 11. It would help the reader to have a dual axis chart with EF also plotted on Figure 8 and SMI also plotted on Figure 11. Though this may make the graphs cluttered and difficult to read.

Figure 10. Please make this figure bigger, it is hard hard to read, and perhaps split into two figures of two graphs each. Please change Tr Th to Tb.Th.

Figures 12 & 13. Why were these 2 specimens omitted?

Figure 13. Please provide one set of axes to aid the reader with orientation. It might also be useful to order the images by age.

Review form: Reviewer 2

Is the manuscript scientifically sound in its present form?

Yes

Are the interpretations and conclusions justified by the results?

Yes

Is the language acceptable?

Yes

Do you have any ethical concerns with this paper?

No

Have you any concerns about statistical analyses in this paper?

No

Recommendation?

Accept with minor revision (please list in comments)

Comments to the Author(s)

In this manuscript, the authors apply their previously proposed Ellipsoid Factor to trabecular bone scans to investigate the relationship between this metric, SMI, and bone volume fraction, and to evaluate this factor as a potential metric to detect bone changes and osteoporosis. I find the work interesting and relevant to the community, especially as it addresses critical issues with the SMI and the correlation with bone volume fraction. The work is well written in general, provides a good amount of information (also supplementary information), and I applaud the authors for the efforts to make EF available to the community through open source tools. However, I do believe that certain aspects should be addressed and discussed more before the manuscript can be accepted for publication. The specific comments are outlined below:

1. In the introduction, the authors address the inherent flaw of the SMI, and mention that “parts of the trabecular bone surface are concave and become smaller when the volume is expanded”. The use of convex and concave is a bit ambiguous in this context. Due to the topological complexity of bone (high genus), it is “on average” hyperbolic, and thus contains a lot of saddle surfaces. For example, the transition between rod and plate regions are locally saddle-shaped. Saddles are both convex and concave, so it is a bit strange to use convexity and concavity as a metric here. The distinction makes sense in 2D (convex and concave curves grow or shrink in response to a parallel offset), but in 3D it is more complex. I would recommend to alter this description, and maybe consider the area change in response to parallel offset ($A(d+t) = A(d)(1 + \langle H \rangle t + \langle K \rangle t^2$), with H and K the mean and Gaussian curvature), see Jinnai et al.: DOI: 10.1016/s8756-3282(01)00672-x

2. The authors focus predominantly on the median, modal, maximum and minimum values of the distributions. While this seems reasonable, I wonder whether it would be possible to consider other metrics for comparing control and disuse distributions, e.g. the Kolmogorov-Smirnov test.

3. The font sizes in the graphs of Figures 9 and 10 should be increased.

4. There should be more discussion on the validity and usefulness of the EF as either a local or global metric. In the discussion chapter, the authors address the volumetric spatial decomposition in rods and plates, as proposed by others in previous years. While the EF, in principle, is also a local metric as it assigns ellipsoids to many points in the trabecular bone volume, the authors primarily use a single scalar to study the differences between different bone specimens (e.g. median or maximum value), effectively providing a “global” picture, like the SMI. A single number is attractive, e.g. for clinical use, but might not be that relevant from a mechanical point of view due to the large heterogeneity of trabecular bone. The authors are therefore recommended to extend the discussion of the local or global nature of the EF, and address whether it could be useful in conjunction with other local analyses, such as the volumetric decomposition in rods and plates.

5. (linked to previous comment). The authors state (line 183) that “very plate- and very rod-like structures co-exist in all samples”, and that (line 281) “a wealth of shapes across the plate-rod continuum exists in any sample”. These are good remarks, highlighting that the “binary” classification into rod-like or plate-like bone is too stringent. However, it would be good if the authors could comment on the use of EF to classify the relative amounts of rod-like or plate-like features within samples. Or maybe the authors even have a different classification in groups, based on their EF (i.e. not based on rod-like or plate-like, but based on the prolateness or oblateness of the ellipsoids).

6. Since the ellipsoid factor is purely based on the ratios of the ellipsoid axes, there is no consideration of the scale of the ellipsoids. In other words, a small rod or a large rod might have the same EF, but different “mechanical” contributions due to their size. The authors are recommended to address the scale consideration in their discussion of the EF.

7. Since the EF algorithm locally fits ellipsoids (and optimizes them), I presume it would also be possible to extract the orientation of the ellipsoids. This could potentially be very interesting, as it could provide much more insight into the mechanical contribution of certain components (e.g. certain features with similar EF magnitude but different orientation will contribute differentially to overall stiffness). The authors are recommended to discuss whether this would be possible and what potential implications would be.

Decision letter (RSOS-201401.R0)

Dear Dr Felder

On behalf of the Editors, we are pleased to inform you that your Manuscript RSOS-201401 "The plate-to-rod transition in trabecular bone loss is elusive" has been accepted for publication in Royal Society Open Science subject to minor revision in accordance with the referees' reports. Please find the referees' comments along with any feedback from the Editors below my signature.

Please submit your revised manuscript and required files (see below) no later than 7 days from today's (ie 02-Mar-2021) date. Note: the ScholarOne system will 'lock' if submission of the revision is attempted 7 or more days after the deadline. If you do not think you will be able to meet this deadline please contact the editorial office immediately.

on behalf of Dr Dirk Drasdo (Associate Editor) and Kevin Padian (Subject Editor)
openscience@royalsociety.org

Associate Editor Comments to Author (Dr Dirk Drasdo):

Comments to the Author:

Dear Authors,

as Royal Society Open Science has a general readership it would be more than appreciated if you could define briefly specialised, technical terms (as for example "disuse osteoporosis"), and could visualise the measures used (for example the EF algorithm, as in Fig. 1 of their reference [14]). This would make your manuscript attractive and readable for non-experts, also for potential cross-inspiration.

With kind regards,
Dirk Drasdo

Reviewer comments to Author:

Reviewer: 1

Comments to the Author(s)

This well-written manuscript presents a well-reasoned, reproducible set of analyses that show the disadvantages of the SMI measure and provides a clear implementation of the superior EF measure of trabecular geometry. Moreover, it adds to the growing body of evidence that the plate-to-rod transition, supposed to occur with trabecular bone loss, is subtle if it exists at all. I would recommend publishing this manuscript after the authors have addressed some minor revisions outlined below using the authors' line numbers. Chief among these in my opinion, are the extent of the non-uniqueness problem visualised in fig S9 and the fact that EF doesn't seem to be completely independent of BV/TV, the major criticism of SMI. I believe minimal addition to the discussion will remedy these problems.

Minor revisions:

L7 & 11: Please be consistent with the terminology and only use remodelling balance or (re)modelling balance.

L14-15: Consider rephrasing this sentence so that the verb is not at the end. E.g. "SMI was designed to estimate how far a trabecular geometry may be considered rod- or plate-like"

L19: Please replace 'vastly' with something to the effect of 'considerable' or 'notable'.

Interspecific differences in BV/TV are often absolutely larger than that measured between osteoporotic and murine controls here, so vastly is subjective as currently used.

L32-39: While I do not disagree with anything in this paragraph I would encourage the authors to highlight the updates made to the EF approach since Salmon et al., 2015. This expansion of earlier work is not simply an application to new osteoporotic data.

L43 & throughout: Here the ellipsoid vector is referred to as an axis, should this not be semi-axis?

L47, Supp. ii & Supp Fig. 9: The non-unique aspect of EF for certain voxels appears to be an issue for the method and should be briefly discussed in the limitations section. In the case presented in the supplementary material, which EF (-1 or 1) is reported for the voxels in the intersections of the two ellipsoids, the first found stochastically by the algorithm? Also, if several runs are performed will the intersecting voxels, with near -1 and near 1 EF values in different runs, be averaged to ~0? Lastly, if the maximal local ellipsoid is 'generally' non-unique then how much of a problem is this for the method? Presumably most trabecular structures will not have exactly the same volume and very different EF as in the extreme example of Fig. S9?

L50-79: This conceptual explanation is clear and well-written

L54-56: Please provide a reference(s) for the disadvantages of distance ridge seeding or briefly explain to the reader why these occur.

L63-64: Please clarify how the third semi-axis is found, presumably two vectors are found in the plane orthogonal to the first?

L64: Please replace 'following' with 'followed'

L71 & throughout: Since ellipsoids are 3D shapes should it not be 'voxel' instead of 'pixel'?

L72-79: I would encourage the authors to explicitly outline the three convergence variables here as otherwise the reader first encounters them, in-text, in the results section.

L86: Why was the median chosen here above mean or mode? The choice of average doesn't appear very important given the distributions depicted but given the mode of EF is significantly related to BV/TV why not report all 3 averages?

L95: Though not necessary, it would be good to know the kernel size of the median filter used for de-noising the images.

L100-101: Please clarify how BV/TV was measured, with Bone[?]

L118-119: In this section, please explain why Tb.Th is measured.

L123: Please change Tr.Th to the standard Tb.Th abbreviation.

L138-141: Please amend the statement concerning the 35 days results in figure 7 to match this reporting.

L143-144: Only correlation was mentioned in the methods section, yet regression appears to be used here. Please be consistent in reporting r or r^2 or explain why one was used instead of the other. Please also report the effect size of the t-tests as the author's have done for the correlation/regression analyses.

L143-144: This result means that EF is related to BV/TV, though to a much lower extent than SMI. As this is a major, and justified, criticism of SMI this result should be addressed in the discussion.

L146: Figure 4 is only referenced in text after figure 8, please consider reordering these figures.

L147: This is the first mention of filling percentage, please explain what this is earlier in the manuscript.

L152: Context around the convergence parameters would help the reader understand why the authors are referring to the final 2 runs of 6 here.

L159: As above, please explain in the methods section why thickness is reported here. Does this implicitly demonstrate plates are not becoming significantly thinner with age?

L161: The r values could be easily confused with the $r^{>2}$ values given in the previous section, please be consistent or more clearly signpost this for the reader.

L165: Please explain why mode is now used as well as the median? Or report all three common averages from the start. Also, the EF to BV/TV association, $r = -0.45$, is described as 'mild' but is still a significant relationship of the same order as -0.65 found for SMI. Perhaps it is better state it is 'lower' than for SMI.

L168: Why were 2 vertebrae not included in figure 13?

L182-184: This observation is very interesting.

L190: Please replace 'support' with 'supports'.

L191: Please replace 'culminates into' with 'culminates in'.

L197-198: Do the authors think local shape variability might also be seen in the murine sample if it were volumetrically larger, i.e. with a higher trabecular number?

L195: Modelling *in silico* and biological trabecular modelling could be confused here, I suggest changing this word. Similarly this is the first reference to 'cancellous' as opposed to 'trabecular' bone, I suggest changing to trabecular bone here for consistency.

L250: As above, why do the authors think that the EF measure was not totally independent of BV/TV, as it was designed to be? I completely agree with the authors that EF appears to be a superior measure of geometry relative to SMI, as it is less dependent on BV/TV, but this is still a result worthy of discussion.

Acknowledgements

L295: This section mentions Python scripts but I cannot find another mention of these in the manuscript. Please clarify or remove.

References

It was not clear initially to me how to access the R/python scripts from reference 27 – perhaps the URL or DOI would be more helpful? https://zenodo.org/record/3556577#.X_NGIfn7SUk

Figures

Figure 1 is intuitive and nicely demonstrates the shape continuum which is key strength of the EF approach. Do all samples conform to this distribution when expressed as a Flinn peak plot, and if so does this tell us anything about the most common shapes found in trabecular bone?

Figure 3. Perhaps add '(purple)' after 'overlap' in the caption

Figure 4. Consider reordering this figure as per above comment. Please also, give an origin and axes for the axis to aid the reader.

Figure 7. See results reporting comment above relating to 35 days.

Figures 8 & 11. It would help the reader to have a dual axis chart with EF also plotted on Figure 8 and SMI also plotted on Figure 11. Though this may make the graphs cluttered and difficult to read.

Figure 10. Please make this figure bigger, it is hard hard to read, and perhaps split into two figures of two graphs each. Please change Tr Th to Tb.Th.

Figures 12 & 13. Why were these 2 specimens omitted?

Figure 13. Please provide one set of axes to aid the reader with orientation. It might also be useful to order the images by age.

Reviewer: 2

Comments to the Author(s)

In this manuscript, the authors apply their previously proposed Ellipsoid Factor to trabecular bone scans to investigate the relationship between this metric, SMI, and bone volume fraction, and to evaluate this factor as a potential metric to detect bone changes and osteoporosis. I find the work interesting and relevant to the community, especially as it addresses critical issues with the SMI and the correlation with bone volume fraction. The work is well written in general, provides

a good amount of information (also supplementary information), and I applaud the authors for the efforts to make EF available to the community through open source tools. However, I do believe that certain aspects should be addressed and discussed more before the manuscript can be accepted for publication. The specific comments are outlined below:

1. In the introduction, the authors address the inherent flaw of the SMI, and mention that “parts of the trabecular bone surface are concave and become smaller when the volume is expanded”. The use of convex and concave is a bit ambiguous in this context. Due to the topological complexity of bone (high genus), it is “on average” hyperbolic, and thus contains a lot of saddle surfaces. For example, the transition between rod and plate regions are locally saddle-shaped. Saddles are both convex and concave, so it is a bit strange to use convexity and concavity as a metric here. The distinction makes sense in 2D (convex and concave curves grow or shrink in response to a parallel offset), but in 3D it is more complex. I would recommend to alter this description, and maybe consider the area change in response to parallel offset ($A(d+t) = A(d)(1 + t + t^2)$, with H and K the mean and Gaussian curvature), see Jinnai et al.: DOI: 10.1016/s8756-3282(01)00672-x
2. The authors focus predominantly on the median, modal, maximum and minimum values of the distributions. While this seems reasonable, I wonder whether it would be possible to consider other metrics for comparing control and disuse distributions, e.g. the Kolmogorov-Smirnov test.
3. The font sizes in the graphs of Figures 9 and 10 should be increased.
4. There should be more discussion on the validity and usefulness of the EF as either a local or global metric. In the discussion chapter, the authors address the volumetric spatial decomposition in rods and plates, as proposed by others in previous years. While the EF, in principle, is also a local metric as it assigns ellipsoids to many points in the trabecular bone volume, the authors primarily use a single scalar to study the differences between different bone specimens (e.g. median or maximum value), effectively providing a “global” picture, like the SMI. A single number is attractive, e.g. for clinical use, but might not be that relevant from a mechanical point of view due to the large heterogeneity of trabecular bone. The authors are therefore recommended to extend the discussion of the local or global nature of the EF, and address whether it could be useful in conjunction with other local analyses, such as the volumetric decomposition in rods and plates.
5. (linked to previous comment). The authors state (line 183) that “very plate- and very rod-like structures co-exist in all samples”, and that (line 281) “a wealth of shapes across the plate-rod continuum exists in any sample”. These are good remarks, highlighting that the “binary” classification into rod-like or plate-like bone is too stringent. However, it would be good if the authors could comment on the use of EF to classify the relative amounts of rod-like or plate-like features within samples. Or maybe the authors even have a different classification in groups, based on their EF (i.e. not based on rod-like or plate-like, but based on the prolateness or oblateness of the ellipsoids).
6. Since the ellipsoid factor is purely based on the ratios of the ellipsoid axes, there is no consideration of the scale of the ellipsoids. In other words, a small rod or a large rod might have the same EF, but different “mechanical” contributions due to their size. The authors are recommended to address the scale consideration in their discussion of the EF.
7. Since the EF algorithm locally fits ellipsoids (and optimizes them), I presume it would also be possible to extract the orientation of the ellipsoids. This could potentially be very interesting, as it could provide much more insight into the mechanical contribution of certain components (e.g. certain features with similar EF magnitude but different orientation will contribute differentially

to overall stiffness). The authors are recommended to discuss whether this would be possible and what potential implications would be.

===PREPARING YOUR MANUSCRIPT===

===PREPARING YOUR REVISION IN SCHOLARONE===

- 1) One version identifying all the changes that have been made (for instance, in coloured highlight, in bold text, or tracked changes);
 - 2) A 'clean' version of the new manuscript that incorporates the changes made, but does not highlight them.
 - An individual file of each figure (EPS or print-quality PDF preferred [either format should be produced directly from original creation package], or original software format).
 - An editable file of each table (.doc, .docx, .xls, .xlsx, or .csv).
 - An editable file of all figure and table captions.
- Note: you may upload the figure, table, and caption files in a single Zip folder.
- Any electronic supplementary material (ESM).
 - If you are requesting a discretionary waiver for the article processing charge, the waiver form must be included at this step.
 - If you are providing image files for potential cover images, please upload these at this step, and inform the editorial office you have done so. You must hold the copyright to any image provided.
 - A copy of your point-by-point response to referees and Editors. This will expedite the preparation of your proof.

- Ensure that your data access statement meets the requirements at <https://royalsociety.org/journals/authors/author-guidelines/#data>. You should ensure that you cite the dataset in your reference list. If you have deposited data etc in the Dryad repository, please only include the 'For publication' link at this stage. You should remove the 'For review' link.
- If you are requesting an article processing charge waiver, you must select the relevant waiver option (if requesting a discretionary waiver, the form should have been uploaded at Step 3 'File upload' above).
- If you have uploaded ESM files, please ensure you follow the guidance at <https://royalsociety.org/journals/authors/author-guidelines/#supplementary-material> to include a suitable title and informative caption. An example of appropriate titling and captioning may be found at [https://figshare.com/articles/Table_S2_from_Is_there_a_trade-off_between_peak_performance_and_performance_breadth_across_temperatures_for_aerobic_sc ope_in_teleost_fishes_/3843624](https://figshare.com/articles/Table_S2_from_Is_there_a_trade-off_between_peak_performance_and_performance_breadth_across_temperatures_for_aerobic_scope_in_teleost_fishes_/3843624).

Author's Response to Decision Letter for (RSOS-201401.R0)

See Appendix A.

Decision letter (RSOS-201401.R1)

Dear Dr Felder,

I am pleased to inform you that your manuscript entitled "The plate-to-rod transition in trabecular bone loss is elusive" is now accepted for publication in Royal Society Open Science.

on behalf of Dr Dirk Drasdo (Associate Editor) and Kevin Padian (Subject Editor)
openscience@royalsociety.org

Appendix A

London, 3rd March 2021

Dear RSOS Team,

We would like to thank the editors and reviewers for their efforts handling our manuscript with the title "The plate-to-rod transition in trabecular bone loss is elusive" submitted to Royal Society Open Science (ID RSOS-201401) in such difficult times. Please find our response to their helpful comments below – we now acknowledge the reviewers' contribution in the manuscript acknowledgements section and have carefully addressed their comments.

Editor

As Royal Society Open Science has a general readership it would be more than appreciated if you could define briefly specialised, technical terms (as for example "disuse osteoporosis"), and could visualise the measures used (for example the EF algorithm, as in Fig. 1 of their reference [14]).

This would make your manuscript attractive and readable for non-experts, also for potential cross-inspiration.

We agree with this. To make our manuscript more accessible to a wider audience, we have defined disuse osteoporosis more explicitly and added an introductory sentence explaining trabecular and cortical bone, and we have improved the caption of Figure 1, which contains all the elements of the figure in [14].

Reviewer 1

This well-written manuscript presents a well-reasoned, reproducible set of analyses that show the disadvantages of the SMI measure and provides a clear implementation of the superior EF measure of trabecular geometry. Moreover, it adds to the growing body of evidence that the plate-to-rod transition, supposed to occur with trabecular bone loss, is subtle if it exists at all. I would recommend publishing this manuscript after the authors have addressed some minor revisions outlined below using the authors' line numbers. Chief among these in my opinion, are the extent of the non-uniqueness problem visualised in fig S9 and the fact that EF doesn't seem to be completely independent of BV/TV, the major criticism of SMI. I believe minimal addition to the discussion will remedy these problems.

Minor revisions:

L7 & 11: Please be consistent with the terminology and only use remodelling balance or (re)modelling balance.

Thanks for this comment. We have gone with "(re)modelling balance".

L14-15: Consider rephrasing this sentence so that the verb is not at the end. E.g. "SMI was designed to estimate how far a trabecular geometry may be considered rod- or plate-like"

L19: Please replace 'vastly' with something to the effect of 'considerable' or 'notable'. Interspecific differences in BV/TV are often absolutely larger than that measured between osteoporotic and murine controls here, so vastly is subjective as currently used.

Thanks. We rephrased both sentences as suggested.

L32-39: While I do not disagree with anything in this paragraph I would encourage the authors to
highlight the updates made to the EF approach since Salmon et al., 2015. This expansion of earlier
work is not simply an application to new osteoporotic data.

We have added two sentences to this effect.

L43 & throughout: Here the ellipsoid vector is referred to as an axis, should this not be semi-axis?

Thanks for pointing this out. We now use semi-axis consistently in the manuscript.

L47, Supp. ii & Supp Fig. 9: The non-unique aspect of EF for certain voxels appears to be an issue
for the method and should be briefly discussed in the limitations section. In the case presented in
the supplementary material, which EF (-1 or 1) is reported for the voxels in the intersections of
the two ellipsoids, the first found stochastically by the algorithm? Also, if several runs are
performed will the intersecting voxels, with near -1 and near 1 EF values in different runs, be
averaged to ~0? Lastly, if the maximal local ellipsoid is 'generally' non-unique then how much of a
problem is this for the method? Presumably most trabecular structures will not have exactly the
same volume and very different EF as in the extreme example of Fig. S9?

Great questions, thank you! We now put EF's non-uniqueness into context in the discussion. We
have also modified the supplementary material in a way that reading it should answer the first two
of the questions. The third and fourth question are handled in the limitations section.

L50-79: This conceptual explanation is clear and well-written

Thank you!

L54-56: Please provide a reference(s) for the disadvantages of distance ridge seeding or briefly
explain to the reader why these occur.

We have reformulated this explanation slightly and added an additional figure in the interest of
clarity.

L63-64: Please clarify how the third semi-axis is found, presumably two vectors are found in the
plane orthogonal to the first?

The semi-axis directions continue to slightly change from iteration to iteration due to the small
random rotations that the ellipsoid experiences as it tries to wiggle into the space provided inside
the trabecular surface. At each iteration, a new growth direction is chosen in the plane defined by
the current (mean) vector from the ellipsoid centre to the set of contact points. We've added a
sentence explaining this.

L64: Please replace 'following' with 'followed'

Typo fixed!

L71 & throughout: Since ellipsoids are 3D shapes should it not be 'voxel' instead of 'pixel'?

We consider the word 'pixel' to be unrelated to the dimensionality of the image, so we have left this
as is. We use 'pixel' to mean a value of any type at a location of any combination of dimensions. This
definition of 'pixel' allows very flexible approaches to image analysis, such as setting the pixel value
to be something complex like the 3x3 ellipsoid tensor, located in 3 spatial dimensions and time. Use
of the word 'voxel' would imply that new names for pixels would be needed for every combination
of value and dimensions.

L72-79: I would encourage the authors to explicitly outline the three convergence variables here
as otherwise the reader first encounters them, in-text, in the results section.

This is a good point, thanks. We have detailed this in the methods section now, under “Assign EF to
each pixel and averaging over runs”.

L86: Why was the median chosen here above mean or mode? The choice of average doesn’t
appear very important given the distributions depicted but given the mode of EF is significantly
related to BV/TV why not report all 3 averages?

As many histograms don’t look very normally distributed, we felt that these descriptive statistics
were more suited than the mean.

L95: Though not necessary, it would be good to know the kernel size of the median filter used for
de-noising the images.

We have added this information in the interest of reproducibility, thanks!

L100-101: Please clarify how BV/TV was measured, with BoneJ?

Bone volume fraction was measured using CTAn, in the unrelated study that kindly provided their
trabecular segmentations for the present study. We specify this now.

L118-119: In this section, please explain why Tb.Th is measured.

We have clarified this by giving our reason at the end of the paragraph in question.

L123: Please change Tr.Th to the standard Tb.Th abbreviation.

Typos fixed!

L138-141: Please amend the statement concerning the 35 days results in figure 7 to match this
reporting.

Thank you for catching this. We’ve double-checked our analyses to make sure we report correctly. It
looks like we reported results in this caption from paired t-tests where the normality assumptions
were not justified. We report this correctly in the statement mentioned here (L138-141). This
mistake does not affect our conclusions.

L143-144: Only correlation was mentioned in the methods section, yet regression appears to be
used here. Please be consistent in reporting r or r^2 or explain why one was used instead of the
other. Please also report the effect size of the t-tests as the authors’ have done for the
correlation/regression analyses.

Thank you for this comment. We now consistently report squared correlation coefficients and clarify
which R function we use to do this. As we understand it at least, it’s not a linear regression analysis
(testing whether a fitted regression slope is significantly different from 0) but rather a test whether
the correlation coefficient is significantly different from 0. So maybe it was our formulation in the
results that was unclear – we have now reformulated this sentence to make clear that we looked at
correlation.

L143-144: This result means that EF is related to BV/TV, though to a much lower extent than SMI.
As this is a major, and justified, criticism of SMI this result should be addressed in the discussion.

We have added a sentence to the second paragraph of the discussion to address this valid point.

L146: Figure 4 is only referenced in text after figure 8, please consider reordering these figures.

We've reordered the text instead, moving the sentence referring to Figure 4 to the start of the
paragraph. This was a simpler solution in our view, as well as a more appropriate one in our view, as
it keeps the histograms and renders close to each other, allowing readers to match the histogram to
the anatomy.

L147: This is the first mention of filling percentage, please explain what this is earlier in the
manuscript.

Thanks, this has been added to the Methods section, under "Assign EF to each pixel and averaging
over runs".

L152: Context around the convergence parameters would help the reader understand why the
authors are referring to the final 2 runs of 6 here.

This is a good suggestion and we have already added some context in the method section based on
an earlier comment. We added a few clarifying words to the discussion now, too.

L159: As above, please explain in the methods section why thickness is reported here. Does this
implicitly demonstrate plates are not becoming significantly thinner with age?

We've added a sentence each to the methods and results section to clarify this. The reviewer is right
in their interpretation of this result.

L161: The r values could be easily confused with the r^2 values given in the previous section, please
be consistent or more clearly signpost this for the reader.

We consistently used squared correlation coefficients now. Thanks for pointing this out.

L165: Please explain why mode is now used as well as the median? Or report all three common
averages from the start. Also, the EF to BV/TV association, $r = -0.45$, is described as 'mild' but is
still a significant relationship of the same order as -0.65 found for SMI. Perhaps it is better state it
is 'lower' than for SMI.

We agree with this and have clarified, additionally noting that the significance (as well as the
strength) of this association is less compelling for EF than for SMI.

L168: Why were 2 vertebrae not included in figure 13?

Purely because 22 images are very awkward to fit onto a page. The two left out had other samples
were chosen randomly, one each from the ones that had bone volume fraction 0.12 and 0.1
respectively, as those were the ones that had the most frequent volume fraction of our samples
(there are three others with BV/TV 0.12 and three others with BV/TV 0.1 which are shown in the
image). The samples were very much part of the analysis and didn't have obvious differences in their
histograms or visual representations. We now clarify this further in the Figure caption – note that
this is the opposite of what often happens in similar studies, where just 1 or 2 "representative
examples" are shown in Figure form.

While double-checking the vertebral renders, we noticed that the renders of sample 14 and 42 were
swapped, which we've now rectified. All the information, and the histograms in the corresponding
figures are correct.

L182-184: This observation is very interesting.

We agree!

L190: Please replace 'support' with 'supports'.

Typo fixed!

L191: Please replace 'culminates into' with 'culminates in'.

Typo fixed!

L197-198: Do the authors think local shape variability might also be seen in the murine sample if
it were volumetrically larger, i.e. with a higher trabecular number?

We assume the reviewer is referring to lines 187-189, which discusses variability of local shape in the
human samples. Beyond the size of the animal and trabecular compartment, the tibia and the
vertebra have a very different shape and mechanical environment, of course, so we would be
cautious about extrapolating variability purely based on volume. It's also probable that mice bred in
experimental facilities don't display the range of variability one would expect from wild samples.
Studying the influence of size on trabecular anatomy is of great interest to us, so we appreciate the
question.

L195: Modelling *in silico* and biological trabecular modelling could be confused here, I suggest
changing this word. Similarly this is the first reference to 'cancellous' as opposed to 'trabecular'
bone, I suggest changing to trabecular bone here for consistency.

We should clarify here that "modelling" here takes a third sense not mentioned by the reviewer in
this context. In this context, we didn't intend *in silico* - we approximately meant "applying
theoretical considerations based on physical laws, reasonable assumptions and structural
engineering principles to calculate mechanical properties using pen and paper" without needing a
computer. We agree with the reviewer that we should be as consistent and clear as we can, so we've
changed both words and added a reference to a seminal book on cellular solids to improve this
sentence.

L250: As above, why do the authors think that the EF measure was not totally independent of
BV/TV, as it was designed to be? I completely agree with the authors that EF appears to be a
superior measure of geometry relative to SMI, as it is less dependent on BV/TV, but this is still a
result worthy of discussion.

While we agree with the reviewer that this result is worthy of discussion, we believe this is already
discussed on L185-187. As we repeatedly make clear in the manuscript, we are not convinced that
such mild associations hovering around a significance level of 0.05 make a good case for a biological
process driving local shape changes, but more investigations may be needed to confirm or deny our
suspicions.

Acknowledgements

L295: This section mentions Python scripts but I cannot find another mention of these in the
manuscript. Please clarify or remove.

This was clarified in the sentence and a reference to the scripts was added. Thank you!

References

It was not clear initially to me how to access the R/python scripts from reference 27 – perhaps the
URL or DOI would be more helpful? https://zenodo.org/record/3556577#.X_NGIfn7SUK

198 **Figures**

Figure 1 is intuitive and nicely demonstrates the shape continuum which is key strength of the EF
approach. Do all samples conform to this distribution when expressed as a Flinn peak plot, and if
so does this tell us anything about the most common shapes found in trabecular bone?

We thank the reviewer for valuing the work that went into this figure, and for this interesting
question. We haven't quantified this, but "eye-balling" it in our samples suggests that yes, the major
peaks roughly follow some hyperbolic-looking curve, and this can skew a bit more towards the
centre or towards the lower left of the plot from sample to sample. However, minor peaks can be
found almost everywhere. I may suggest that the typical EF=0 shape is not a sphere at all, but rather
a "surfboard" with semi-axis lengths at a ratio of something like 1:2:4 or 1:3:9. This a possible
avenue of further investigation, as suggested in our discussion on L268-274 of the submitted
manuscript.

Figure 3. Perhaps add '(purple)' after 'overlap' in the caption

Good idea. Done.

Figure 4. Consider reordering this figure as per above comment. Please also, give an origin and
axes for the axis to aid the reader.

Re-ordering has been handled as described in the above comment. We've added appropriate axes to
match the description in the caption.

Figure 7. See results reporting comment above relating to 35 days.

See response to comment pertaining to L138-141.

Figures 8 & 11. It would help the reader to have a dual axis chart with EF also plotted on Figure 8
and SMI also plotted on Figure 11. Though this may make the graphs cluttered and difficult to
read.

We have left this as is, because we feel the amount of information in Fig 11 is more than enough to
digest as a reader already (as the reviewer also suggests).

Figure 10. Please make this figure bigger, it is hard hard to read, and perhaps split into two
figures of two graphs each. Please change Tr Th to Tb.Th.

We have fixed this typo and doubled height and width of each subfigure.

Figures 12 & 13. Why were these 2 specimens omitted?

See answer to comment on L168.

Figure 13. Please provide one set of axes to aid the reader with orientation. It might also be
useful to order the images by age.

We decided to order by bone volume fraction rather than age as the relationship between bone
volume fraction and other trabecular parameters is central to our study (although we agree with the
reviewer that age would also be a possibility). We have added a sentence to the caption to aid with
orientation.

Reviewer: 2

Comments to the Author(s)

In this manuscript, the authors apply their previously proposed Ellipsoid Factor to trabecular bone
scans to investigate the relationship between this metric, SMI, and bone volume fraction, and to
evaluate this factor as a potential metric to detect bone changes and osteoporosis. I find the work
interesting and relevant to the community, especially as it addresses critical issues with the SMI and
the correlation with bone volume fraction. The work is well written in general, provides a good
amount of information (also supplementary information), and I applaud the authors for the efforts
to make EF available to the community through open source tools. However, I do believe that
certain aspects should be addressed and discussed more before the manuscript can be accepted for
publication. The specific comments are outlined below:

1. In the introduction, the authors address the inherent flaw of the SMI, and mention that “parts
of the trabecular bone surface are concave and become smaller when the volume is expanded”.
The use of convex and concave is a bit ambiguous in this context. Due to the topological
complexity of bone (high genus), it is “on average” hyperbolic, and thus contains a lot of saddle
surfaces. For example, the transition between rod and plate regions are locally saddle-shaped.
Saddles are both convex and concave, so it is a bit strange to use convexity and concavity as a
metric here. The distinction makes sense in 2D (convex and concave curves grow or shrink in
response to a parallel offset), but in 3D it is more complex. I would recommend to alter this
description, and maybe consider the area change in response to parallel offset ($A(d+t) = A(d) \cdot (1 +$
$t + t^2)$, with H and K the mean and Gaussian curvature), see Jinnai et al.: DOI: 10.1016/s8756-
3282(01)00672-x

Thanks for making this important point and for making us aware of this interesting reference. We
have adapted our explanation to reflect this in the introduction and the discussion.

2. The authors focus predominantly on the median, modal, maximum and minimum values of the
distributions. While this seems reasonable, I wonder whether it would be possible to consider
other metrics for comparing control and disuse distributions, e.g. the Kolmogorov-Smirnov test.

We indeed toyed with the idea of using the Kolmogorov-Smirnov test for this purpose. This might be
interesting to pursue in the future. We might need even more complicated statistics for this in the
future, as we’d need the analysis to compare (possibly two-dimensional? See Flinn plots!)
distributions while accounting for sample and group.

3. The font sizes in the graphs of Figures 9 and 10 should be increased.

We have addressed this by rearranging the subfigures to take up a larger amount of space.

4. There should be more discussion on the validity and usefulness of the EF as either a local or
global metric. In the discussion chapter, the authors address the volumetric spatial
decomposition in rods and plates, as proposed by others in previous years. While the EF, in
principle, is also a local metric as it assigns ellipsoids to many points in the trabecular bone
volume, the authors primarily use a single scalar to study the differences between different bone
specimens (e.g. median or maximum value), effectively providing a “global” picture, like the SMI.

A single number is attractive, e.g. for clinical use, but might not be that relevant from a
mechanical point of view due to the large heterogeneity of trabecular bone. The authors are
therefore recommended to extend the discussion of the local or global nature of the EF, and
address whether it could be useful in conjunction with other local analyses, such as the
volumetric decomposition in rods and plates.

5. (linked to previous comment). The authors state (line 183) that “very plate- and very rod-like
structures co-exist in all samples”, and that (line 281) “a wealth of shapes across the plate-rod
continuum exists in any sample”. These are good remarks, highlighting that the “binary”
classification into rod-like or plate-like bone is too stringent. However, it would be good if the
authors could comment on the use of EF to classify the relative amounts of rod-like or plate-like
features within samples. Or maybe the authors even have a different classification in groups,
based on their EF (i.e. not based on rod-like or plate-like, but based on the prolateness or
oblateness of the ellipsoids).

We thank the reviewer for comments 4 and 5, which give us an opportunity to expand our
discussion, which we have taken.

6. Since the ellipsoid factor is purely based on the ratios of the ellipsoid axes, there is no
consideration of the scale of the ellipsoids. In other words, a small rod or a large rod might have
the same EF, but different “mechanical” contributions due to their size. The authors are
recommended to address the scale consideration in their discussion of the EF.

7. Since the EF algorithm locally fits ellipsoids (and optimizes them), I presume it would also be
possible to extract the orientation of the ellipsoids. This could potentially be very interesting, as it
could provide much more insight into the mechanical contribution of certain components (e.g.
certain features with similar EF magnitude but different orientation will contribute differentially
to overall stiffness). The authors are recommended to discuss whether this would be possible and
what potential implications would be.

We now address the important observations in 6. and 7. under “Limitations and Future work” in the
Discussion. Thanks!
